# SoundMorpher: Perceptually-Uniform Sound Morphing with Diffusion Model

## Abstract

We present SoundMorpher, a sound morphing method that generates perceptually uniform morphing trajectories using a diffusion model. Traditional sound morphing methods models the intractable relationship between morph factor and perception of the stimuli for resulting sounds under a linear assumption, which oversimplifies the complex nature of sound perception and limits their morph quality. In contrast, SoundMorpher explores an explicit proportional mapping between the morph factor and the perceptual stimuli of morphed sounds based on Mel-spectrogram. This approach enables smoother transitions between intermediate sounds and ensures perceptually consistent transformations, which can be easily extended to diverse sound morphing tasks. Furthermore, we present a set of quantitative metrics to comprehensively assess sound morphing systems based on three objective criteria, namely, correspondence, perceptual intermediateness, and smoothness. We provide extensive experiments to demonstrate the effectiveness and versatility of SoundMorpher in real-world scenarios, highlighting its potential impact on various applications such as creative music composition, film post-production and interactive audio technologies [1].

## 1 Introduction

Sound morphing is a technique to create a seamless transformation between multiple sound recordings. The goal is to produce perceptual intermediate sounds that gradually change from one sound to another. Sound morphing has a wide range of applications, including music compositions, synthesizers, psychoacoustic experiments to study timbre spaces (Caetano & Rodet, 2011; Hyrkas, 2021), and practical applications such as film post-production, AR or VR interactive games, and adaptive audio content in video games (Qamar et al., 2020; Siddiq, 2015).

Traditional sound morphing methods used the interpolation principle of sound synthesis technique, which relies on interpolating the parameters of a sinusoidal model (Tellman et al., 1995; Osaka, 1995; Williams et al., 2014). Others make use of digital signal processing techniques to explore high-level audio features in the time-frequency domain to achieve more effective and continuous morphing (Williams et al., 2014; Brookes & Williams, 2010; Caetano & Rodet, 2010; 2011; Roma et al., 2020; Caetano, 2019). However, these methods are limited to applications such as producing inharmonic and noisy environmental sounds (Gupta et al., 2023; Kamath et al., 2024). Despite the increasing interest in applying machine learning to sound generation, there has only been limited exploration in sound morphing. Recent approaches (Zou et al., 2021; Gupta et al., 2023; Kim et al., 2019b; Kamath et al., 2024) have shown their superior effectiveness compared to traditional methods in various scenarios. However, we observed several critical limitations of those existing methods. Firstly, they are primarily designed for static or cyclostationary morphing (see Sec. 3.1), limiting their applicability to dynamic sound transformations. Secondly, these approaches often lack sufficient quantitative evaluation, limiting further analysis of their effectiveness. Thirdly, they require training on task-specific datasets, which limits their application in different scenarios. Most importantly, they typically assume a linear relationship between morphing factors and sound perception, and achieve smooth morphing by gradually changing the morph factor. This assumption oversimplifies the complex nature of sound perception, as gradually changing morph factors does

---

[1]Our demonstration for listening is in the supplementary material.

not inherently result in smooth perceptual transitions. To this end, our goal is to develop a method that achieves perceptually coherent morphing, ensuring seamless and natural sound transition.

In this paper, we introduce SoundMorpher, a sound morphing method that produces perceptually smooth and intermediate morphing, comprising the following key contributions.

- SoundMopher is the first open-world sound morphing method based on a pre-trained diffusion model, which integrates typical morph tasks such as static, dynamic and cyclostationary morphing. Unlike prior works (Kim et al., 2019b; Gupta et al., 2023), SoundMorpher can be broadly applied to various real-world tasks without requiring extensive retraining.
- We propose the sound perceptual distance proportion (SPDP), which explicitly connects morph factors and perceptual stimuli of morphed results. This allows SoundMorpher to produce morphing paths with a uniform change in perceptual stimuli, achieving more seamless perceptual transitions compared to existing methods (Kamath et al., 2024).
- We adapt a set of comprehensive quantitative metrics according to criteria proposed by Caetano & Osaka (2012) for evaluation, addressing the lack of quantitative assessment for sound morphing systems (Caetano, 2019; Zou et al., 2021; Caetano & Rodet, 2013) and may offer insights for analyzing and comparing future sound morphing methods.
- We provide extensive experiments to demonstrate that SoundMorpher can be effectively applied to several potential applications in broader real-world scenarios, including musical instrument timbre morphing, music morphing and environmental sound morphing.

## 2 RELATED WORK

In this section, we first present a detailed review of related works on sound morphing task. Then, we also briefly introduce tasks that are similar to sound morphing and clarify the differences.

**Sound morphing.** Traditional sound morphing methods rely on interpolating parameters of a sinusoidal sound synthesis model (Tellman et al., 1995; Osaka, 1995; Williams et al., 2014; Primavera et al., 2012). To achieve more effective and continuous morphing, Williams et al. (2014); Brookes & Williams (2010); Caetano & Rodet (2010; 2011); Roma et al. (2020); Caetano (2011) target on exploring perceptual spectral domain audio features by digital signal processing techniques, such as MFCCs, spectral envelope, etc.. Others such as Kazazis et al. (2016) involve a hybrid approach that extracts audio descriptors to morph accordingly and interpolate between the spectrotemporal fine structures of two endpoints according to morph factors. Machine learning sound morphing methods offer advantages such as high morphing quality by leveraging semantic representation interpolation within a model instead of traditional audio feature interpolation. Zou et al. (2021) proposes a non-parallel many-to-one static timbre morphing framework that integrates and fine-tunes the machine learning technique (i.e., DDSP-autoencoder (Engel et al., 2020)) with spectral feature interpolation (Caetano & Rodet, 2013). Kim et al. (2019b) targets synthesizing music corresponding to a note sequence and timbre, which uses non-linear instrument embedding as timbre control parameters under a pretrained WaveNet (Engel et al., 2017) to achieve timbre morphing between instruments. Luo et al. (2019) learns latent distributions of VAEs to disentangle representations for pitch and timbre of musical instrument sounds. Tan et al. (2020) uses a GM-VAE to achieve style morphing to generate realistic piano performances in the audio domain following temporal style conditions for piano performances, which morphs the conditions such as onset roll and MIDI note into input audio. MorphGAN (Gupta et al., 2023) targets on audio texture morphing by interpolating within conditional parameters, and trained the model on a water-wind texture dataset. A recent concurrent work by Kamath et al. (2024) uses a pre-trained AudioLDM (Liu et al., 2023) to morph sound between two text prompts. In contrast, we focus on classical sound morphing, where the morphing process is performed directly between two given audios rather than between text prompts. A key advantage of our method is its ability to provide precise guidance during the morphing process, as the target audio delivers exact information on how the source sound should evolve—something that text prompts cannot always achieve, for example, morphing between two music compositions.

**Synthesizer preset interpolation.** Synthesizer preset interpolation achieves sound morphing by developing models that compute interpolations within the domain of synthesis parameters for a black-box synthesizer (Le Vaillant & Dutoit, 2023; Dutoit et al., 2023; Le Vaillant & Dutoit, 2024). Unlike classical sound morphing, which perceptually blends two audio files into an intermediate sound, synthesizer preset interpolation treats the synthesizer as a non-differentiable black box, with

presets composed of both numerical and categorical parameters. By smoothly interpolating between these presets, the task aims to achieve seamless morphing of synthesized sounds.

**Text-to-audio editing.** Text-to-audio editing is the process of using text queries to edit audio. With the success of diffusion models in image editing tasks, recent works target zero-shot audio editing with text instructions (Manor & Michaeli, 2024; Zhang et al., 2024; Lan et al., 2024) involving tasks such as inpainting, outpainting, timbre transfer, music genre transfer, or vocals removal.

**Timbre transfer.** Timbre transfer is a specific task that aims at converting the sound of a musical piece by one instrument (source) into the same piece played by another instrument (target). This concerns the task of converting a musical piece from one timbre to another while preserving the other music-related characteristics (Comanducci et al., 2024; Jain et al., 2020; Li et al., 2024).

**Voice conversion and morphing.** Voice conversion (VC) involves modifying vocal characteristics of a source speech to match a target speaker, either by using target speeches or text (Li et al., 2023; Yao et al., 2024; Niu et al., 2024; Sheng et al., 2024). The primary objective of VC is to alter the vocal identity to closely resemble the target voice style, while preserving the linguistic content of the source speech. Voice morphing is a broader scope, focusing on blending or transforming one voice into another. This often involves creating an intermediate voice that incorporates characteristics of both source and target voices, allowing for gradual transitions between them (Sheng et al., 2024).

# 3 PRELIMINARIES

## 3.1 SOUND MORPHING

Sound morphing aims to produce intermediate sounds as different combinations of model source sound $\hat{S}_1$ and target sound $\hat{S}_2$ (Caetano & Rodet, 2010; 2011), which can be formulated as

$$M(\alpha, t) = (1 - \alpha(t))\hat{S}_1 + \alpha(t)\hat{S}_2 \tag{1}$$

Each step is characterized by one value of a single parameter $\alpha$, the so-called morph factor, which ranges between 0 and 1, where $\alpha = 0$ and $\alpha = 1$ produce resynthesized source and target sounds, respectively. Due to the intrinsic temporal nature of sounds, sound morphing usually involve three main types: *dynamic morphing*, where $\alpha$ gradually transfers from 0 to 1 over time dimension (Kazazis et al., 2016), *static morphing*, where a single morph factor $\alpha$ leads to an intermediate sound between source and target (Sethares & Bucklew, 2015), and *cyclostationary morphing* where several hybrid sounds are produced in different intermediate points (Slaney et al., 1996).

To solve the limitation on previous works that target on expensive perceptual evaluation only, Caetano & Osaka (2012) proposes three objective criteria for sound morphing techniques: (1) *Correspondence.* The morph is achieved by a description whose elements are intermediate between source and target sounds, highlighting semantic level transition; (2) *Intermediateness.* The morphed objects should be perceived as intermediate between source and target sounds, evaluating perceptual level correlation; (3) *Smoothness.* The morphed sounds should change gradually (i.e., 'smoothly') from source to target sounds, by the same amount of perception increment. Under the assumption of linear perceptual stimuli, adding the same factor should increase the same amount of perception. In this study, we evaluate SoundMorpher according to the three criteria by a series of comprehensive objective quantitative metrics in addition to perceptual evaluation.

## 3.2 LATENT DIFFUSION MODEL ON AUDIO GENERATION

SoundMorpher utilizes a pretrained text-to-audio (TTA) latent diffusion model (LDM) (Rombach et al., 2022) to achieve sound morphing. This approach offers the advantage of performing various types of sound morphing without the need to train the entire model or use additional datasets. Specifically, we use AudioLDM2 (Liu et al., 2024), a multi-modality conditions to audio model. It employs a pre-trained variational autoencoder (VAE) (Kingma & Welling, 2013) to compress audio $x$ into a low-dimension latent space as VAE representations $z$. AudioLDM2 generates latent variables $z_0$ from a Gaussian noise $z_T$ given the condition $C$ and further reconstruct audio $\hat{x}$ from $z_0$ by VAE decoder and a vocoder (Kong et al., 2020). AudioLDM2 uses an intermediate feature $Y$ as an abstraction of audio data $x$ to bridge the gap between conditions $C$ and audio $x$, named language of audio (LOA). The LOA feature is obtained by a AudioMAE (Huang et al., 2022; Tan et al., 2024) and a series of post-processing formulated as $Y = \mathcal{A}(x)$. The generation function $\mathcal{G}(\cdot)$ is achieved

by a LDM. In the inference phase, AudioLDM2 approximates LOA feature by the given condition as $\hat{Y} = \mathcal{M}(C)$ using a fine-tuned GPT-2 model (Radford et al., 2019). Then generates audios conditioned on the estimated LOA feature $\hat{Y}$ and an extra text embedding $E_{T5}$ from a FLAN-T5 (Chung et al., 2024) with a LDM as $\hat{x} = \mathcal{G}(\hat{Y}, E_{T5})$. We denote the conditional embeddings in AudioLDM2 as $E = \{\hat{Y}, E_{T5}\}$, therefore, the generative process becomes $\hat{x} = \mathcal{G}(E)$.

**Diffusion Models.** The LDM performs a forward diffusion process during training, which is defined as a Markov chain that gradually adds noise to the VAE representation $z_0$ over $T$ steps as $z_t = \sqrt{1 - \beta_t} z_{t-1} + \sqrt{\beta_t} \epsilon_t$. where $\epsilon_t \sim N(0, I)$ and noise schedule hyperparameter $\beta_t \in [0, 1]$. Therefore, we can derive the distribution of $z_t$ given $z_0$ as $q(z_t | z_0) = \sqrt{\gamma_t} z_0 + \sqrt{1 - \gamma_t} \epsilon_t$, where $\gamma_t = \prod_{t=1}^{t} 1 - \beta_t$. The LDM learns a backward transition $\epsilon_\theta(z_t, t)$ from the prior distribution $N(0, I)$ to the data distribution $z$, that predicts the added noise $\epsilon_t$ (Ho et al., 2020). Following the objective function of denoising diffusion probabilistic models (DDPM) (Ho et al., 2020), the objective function for training AudioLDM2 is

$$\min_\theta \mathcal{L}_{DPM} = \text{argmin}_\theta[\mathbb{E}_{z_0, E, t \sim \{1, ..., T\}} ||\epsilon_\theta(z_t, E, t) - \epsilon_t||_2^2] \tag{2}$$

To reduce computational demands on inference, AudioLDM2 uses denoising diffusion explicit models (DDIM) (Song et al., 2020), which provides an alternative solution and enables significantly reduced sampling steps with high generation quality. The DDIM reverse diffusion process is

$$z_{t-1} = \sqrt{\gamma_{t-1}}(\frac{z_t - \sqrt{1 - \gamma_t} \epsilon_\theta(z_t, E, t)}{\sqrt{\gamma_t}}) + \sqrt{1 - \gamma_{t-1} - \sigma_t^2} \epsilon_\theta(z_t, E, t) + \sigma_t \epsilon_t \tag{3}$$

We can revise a deterministic mapping between $z_0$ and its latent state $z_T$ once the model is trained (Dhariwal & Nichol, 2021; Yang et al., 2023) by the following equation

$$\frac{z_{t+1}}{\sqrt{\gamma_{t+1}}} - \frac{z_t}{\sqrt{\gamma_t}} = (\sqrt{\frac{1 - \gamma_{t+1}}{\gamma_{t+1}}} - \sqrt{\frac{1 - \gamma_t}{\gamma_t}})\epsilon_\theta(z_t, E, t) \tag{4}$$

## 4 METHOD

Given a source and target audio pair $\{x^{(0)}, x^{(1)}\}$, sound morphing aims to generate intermediate sounds $x^{(\alpha(t))}$ between the audio pair given morph factors $\alpha \in [0, 1]$. To account for the variation of $\alpha$ over time in Equation 1, we discretize the function $\alpha(t)$ where $t \in [0, T]$ into $N$ elements, resulting in a morphed sequence of sounds $\{x^{(\alpha_i)}\}_{i=1}^{N}$ based on $\{\alpha_i\}_{i=1}^{N}$. According to the smoothness criteria proposed by Caetano & Osaka (2012), the desired sound morphing technique should have *smooth linear perceptual stimuli* when the morph factor $\alpha$ varies in the sequence $\{\alpha_i\}_{i=1}^{N}$. Therefore, we define $p_i$ to represent the perceptual stimuli of the morphed audio $x^{(\alpha_i)}$ given morph factor $\alpha_i$. However, the relationship $\mathcal{P}(\cdot)$ between morph factor $\alpha$ and perceptual stimuli $p$ is intractable. Our goal is to find a discrete morph factor sequence $\{\alpha_i\}_{i=1}^{N}$ such that for each transition, the perceptual stimuli difference $\Delta p$ is a constant value. Therefore, we formulate the problem as

$$p_{i+1} - p_i \equiv \mathcal{P}(x^{(\alpha_{i+1})}) - \mathcal{P}(x^{(\alpha_i)}) = \Delta p, \ i \in [1, ..., N - 1] \tag{5}$$

This formulation is a refined sound morphing problem where, rather than controlling morph factor $\alpha$, we control the constant perceptual stimuli difference $\Delta p$ to find the optimal trajectory with morph factors $\{\alpha_i\}_{i=1}^{N}$ that will achieve *perceptually uniform sound morphing* [2].

In Section 4.1 we introduce feature interpolation and model adaption with a pre-trained AudioLDM2. This method allows high-quality intermediate morph results to be obtained by controlling morph factor $\alpha$. To achieve perceptually uniform sound morphing as in Equation 5, we explore an explicit connection $\mathcal{P}(\cdot)$ between perceptual stimuli $p$ and morph factor $\alpha$ in Section 4.2. In Section 4.3, we provide extensions of our method on the different morphing methods discussed in Section 3.1 to show the advantages of perceptually uniform sound morphing.

### 4.1 FEATURE INTERPOLATION AND MODEL ADAPTION

**Interpolating optimized conditional embeddings.** We first introduce text-guided conditional embedding optimization strategy under a pre-trained AudioLDM2, which retrieves corresponding conditional embeddings $E$ of the given audio data. As mentioned in Section 3.2, AudioLDM2 accepts

---

[2]See Appendix 7.1 for overall SoundMorpher pseudo algorithm pipeline and further implementation details.

two conditional inputs: LOA feature $Y$ and text embedding $E_{T5}$. We denote $E = \{Y, E_{T5}\}$ as the overall conditional embedding inputs for AudioLDM2. The LOA feature $Y$ is an abstraction of audio data which is semantically structured, and $E_{T5}$ captures sentence-level of representations. To retrieve corresponding conditional embeddings of the given audio data, we first obtain the latent variables $z_0^{(0)}$ and $z_0^{(1)}$ of audio $x^{(0)}$ and $x^{(1)}$ from the pre-trained VAE in AudioLDM2 pipeline. We initialize a simple common text prompt (e.g., '*An audio clip of sound*') as a text guidance condition $C$ to obtain $E$ by GPT-2 encoder and FLAN-T5 encoder in AudioLDM2 pipeline, respectively, as $E^{(0)}$ and $E^{(1)}$. Instead of optimizing the model parameters, we freeze the model parameters and optimize the conditional embedding $E^{(0)}$ and $E^{(1)}$ by the denoising objective function in Equation 2

$$E^{(0)} = \arg\min_E \mathcal{L}_{DPM}(z_0^{(0)}, E; \theta) \text{ and } E^{(1)} = \arg\min_E \mathcal{L}_{DPM}(z_0^{(1)}, E; \theta) \tag{6}$$

The optimized conditional embeddings $E^{(0)}$ and $E^{(1)}$ fully encapsulate the abstract details of audios $x^{(0)}$ and $x^{(1)}$. Due to the semantically structured nature of the conditional embeddings, the conditional distributions $p_\theta(z|E^{(0)})$ and $p_\theta(z|E^{(1)})$ closely mirror the degree of audio variation between the audio pair. To explore the data distribution that conceptually intermediate between $z^{(0)}$ and $z^{(1)}$, we bridge these two distributions through linear interpolation. Specifically, we define the interpolated conditional distribution as $p_\theta(z|E^{(\alpha)}) := p_\theta(z|(1-\alpha)E^{(0)} + \alpha E^{(1)})$, where $\alpha \in [0, 1]$.

**Interpolating latent state.** The conditional embedding represents the conceptual abstract of audio data. However, we also wish to smoothly morph the content of the audio pair. Following Song et al. (2020) and Yang et al. (2023), we smoothly interpolate between $z_0^{(0)}$ and $z_0^{(1)}$ by spherical linear interpolation (slerp) to their starting noise $z_T^{(0)}$ and $z_T^{(1)}$ and further obtained the interpolated latent state $z_T^{(\alpha)} := \frac{\sin(1-\alpha)\omega}{\sin\omega} z_T^{(0)} + \frac{\sin\alpha\omega}{\sin\omega} z_T^{(1)}$, where $\omega = \arccos(\frac{z_T^{(0)\top} z_T^{(1)}}{||z_T^{(0)}||||z_T^{(1)}||})$. The denoised latent variable $z_0^{(\alpha)}$ is obtained by applying a diffusion denoising process on the interpolated starting noise $z_T^{(\alpha)}$ and conditioning on the interpolated conditional embedding $E^{(\alpha)}$. The final morphed audio result $x^{(\alpha)}$ is obtained from $z_0^{(\alpha)}$ by the VAE decoder and a vocoder.

**Model adaption.** Model adaptation helps to limit the degree of morphed variation by suppressing high-density regions that not related to the given inputs (Yang et al., 2023). We use LoRA (Hu et al., 2021) to inject a small amount of trainable parameters for efficient model adaptation. We fine-tune AudioLDM2 with LoRA trainable parameters using $z^{(0)}$ and $z^{(1)}$. See Appendix 7.2 for details.

## 4.2 PERCEPTUALLY UNIFORM SOUND MORPHING

**Sound perceptual distance proportion (SPDP).** The relationship between morph factor $\alpha$ and perceptual stimuli $p$ is intractable. Our goal is to establish an objective quantitative metric that links $p_i$ and $x^{(\alpha_i)}$ as in Equation 5. This metric should satisfy two key conditions: (1) the output $p$ should increase monotonically as $\alpha$ increases; (2) it should accurately represent perceptual differences between $x^{(\alpha)}$ and $\{x^{(0)}, x^{(1)}\}$, ensuring a smooth transition through intermediate states. Therefore, we propose the *sound perceptual distance proportion* between $x^{(\alpha)}$ and $\{x^{(0)}, x^{(1)}\}$. We define $p_i \in \mathbb{R}^2$ as a 2D vector to represent the perceptual proximity of $x^{(\alpha_i)}$ to both $x^{(0)}$ and $x^{(1)}$. Instead of extracting numerous audio features through traditional signal processing techniques, we use Mel-scaled spectrogram to capture perceptual and semantic information on audio. Mel-spectrogram (Tzanetakis & Cook, 2002) provides a pseudo-3D representation of audio signals, with one axis representing time and the other representing frequency on the Mel scale (Stevens et al., 1937), while the values denote the magnitude of each frequency at specific time points. The advantage of using Mel-spectrogram lies in the Mel filter banks, which map frequencies to equal pitch distances that correspond to how humans perceive sound (Sturm, 2013; Müller, 2015). Denoting $x_{mel}^{(\alpha_i)}$ as the Mel-spectrogram of audio $x^{(\alpha_i)}$, the SPDP $p_i$ between two endpoint audios $x^{(0)}$ and $x^{(1)}$ given $\alpha_i$ is defined as

$$p_i = [\frac{||x_{mel}^{(\alpha_i)} - x_{mel}^{(0)}||_2}{||x_{mel}^{(\alpha_i)} - x_{mel}^{(0)}||_2 + ||x_{mel}^{(\alpha_i)} - x_{mel}^{(1)}||_2}, \frac{||x_{mel}^{(\alpha_i)} - x_{mel}^{(1)}||_2}{||x_{mel}^{(\alpha_i)} - x_{mel}^{(0)}||_2 + ||x_{mel}^{(\alpha_i)} - x_{mel}^{(1)}||_2}] \tag{7}$$

**Binary search with constant SPDP increment.** To produce a perceptually smooth morphing trajectory with a constant perceptual stimuli increment, we use binary search to seek the corresponding

Table 1: Timbral morphing for musical instruments compared to baseline on different instruments.

| Group | Method | FAD ↓ | FD ↓ | $\text{CDPAM}_T$ ↓ | $\text{CDPAM}_{mean \pm std}$ ↓ | $\mathcal{L}_2^{timbre}$ ↓ | $\text{CDPAM}_E$ ↓ |
|---|---|---|---|---|---|---|---|
| Piano ↔ Guitar | SMT | 24.73 | 102.57 | 1.170 | $0.116 \pm 0.074$ | 1.263 | 0.122 |
| | Ours | 5.21 | 41.11 | 0.404 | $0.044 \pm 0.020$ | 0.466 | 0.132 |
| Harp ↔ Kalimaba | SMT | 13.46 | 88.89 | 1.495 | $0.150 \pm 0.117$ | 1.355 | 0.182 |
| | Ours | 4.67 | 37.92 | 0.768 | $0.076 \pm 0.089$ | 0.462 | 0.159 |
| Taiko ↔ Hihat | SMT | 8.51 | 131.57 | 2.339 | $0.234 \pm 0.332$ | 1.584 | 0.732 |
| | Ours | 3.32 | 47.59 | 1.314 | $0.131 \pm 0.058$ | 0.359 | 0.102 |
| Piano ↔ Violin | SMT | 21.38 | 90.63 | 1.902 | $0.190 \pm 0.069$ | 0.558 | 0.217 |
| | Ours | 3.42 | 20.14 | 0.782 | $0.078 \pm 0.020$ | 0.415 | 0.085 |
| Piano ↔ Organ | SMT | 21.36 | 63.26 | 1.291 | $0.129 \pm 0.074$ | 1.106 | 0.097 |
| | Ours | 3.29 | 19.73 | 0.233 | $0.023 \pm 0.010$ | 0.423 | 0.097 |

$\{\alpha_i\}_{i=1}^N$ based on a constant $\Delta p$. The target SPDP sequence $\{p_i\}_{i=1}^N$ is obtained by an interpolation $p_i = (1 - \frac{i-1}{N-1})p^{(0)} + \frac{i-1}{N-1}p^{(1)}$, where the two endpoints are $p^{(0)} = [0, 1]^T$ and $p^{(1)} = [1, 0]^T$. See Algorithm 2 in Appendix 7.3 for detail pseudo algorithm.

### 4.3 Controllable Sound Morphing with Discrete $\alpha$ Series

By controlling the discrete morph factor sequence $\{\alpha_i\}_{i=1}^N$ to produce a morphed series $\{x^{(\alpha_i)}\}_{i=1}^N$ with constant $\Delta p$, we achieve three typical morphing methods as follows.

**Static morphing.** To achieve controllable static morphing, we control the target SPDP point $p$, which represents how the desired output perceptually intermediate between $x^{(0)}$ and $x^{(1)}$. We find the corresponding $\alpha$ value by the binary search with the target $p$ and further obtain a morphed result $x^{(\alpha)}$. Pseudocodes for static morphing are in Algorithm 3.

**Cyclostationary morphing.** To produce $N$ perceptually uniform hybrid sounds between $x^{(0)}$ and $x^{(1)}$, we first obtain $N$ uniform interpolated SPDP points $\{p_i\}_{i=1}^N$. Then we find corresponding morph factors $\{\alpha_i\}_{i=1}^N$ and further obtain $N$ morphed results $\{x^{(\alpha_i)}\}_{i=1}^N$ as in Algorithm 4.

**Dynamic morphing.** Dynamic morphing performs sound morphing over time, but one challenge is that if the morphing path fails to ensure perceptual intermediateness and content correspondence, the resulting sounds may exhibit perceptual discontinuities or unnatural intermediate stages. As in Algorithm 5, we obtain $N$ interpolated target SPDP points $\{p_i\}_{i=1}^N$ with $\Delta p$. The corresponding morph factors $\{\alpha_i\}_{i=1}^N$ are determined by binary search with the target SPDP points. Each morphed result $x^{(\alpha_i)}$ contributes a segment of duration $\frac{T}{N}$, producing an audio segment $\tilde{x}^{(\alpha_i)}$ according to index $i$. The final audio signal is obtained by concatenating these morphed segments, resulting in

$$[x_0, x_1, ..., x_T] = \text{concat}(\tilde{x}^{(0)}, \tilde{x}^{(\alpha_1)}, ..., \tilde{x}^{(1)}) \tag{8}$$

## 5 Experiment

In this section, we showcase three applications of SoundMorpher in real-world scenarios: *Timbral morphing for musical instruments*, *Environmental sound morphing*, and *Music morphing*.

### 5.1 Evaluation Metric

We verify SoundMorpher according to the criteria mentioned in Section 3.1. **Correspondence.** We design a metric that computes absolute error for the mid-point MFCCs proportion, namely $\text{MFCCs}_{\mathcal{E}}$, for description correspondence (see Appendix 8.2 for detail). We use *Fréchet audio distance* (FAD) (Kilgour et al., 2018) and *Fréchet distance* (FD) (Eiter & Mannila, 1994) between morphed audios and sourced audios to verify semantic similarity and morphed audio quality (see Appendix 8.3 for detail). **Intermediateness.** We use total CDPAM (Manocha et al., 2021) by $\text{CDPAM}_T = \sum_{i=1}^{N-1} \text{CDPAM}(x^{(\alpha_i)}, x^{(\alpha_{i+1})})$ for morph sequence to reflect direct perceptual intermediateness. A smaller $\text{CDPAM}_T$ indicates the morphing sequence exhibits higher perceptual intermediate similarity between consecutive sounds, suggesting intermediate consistency. **Smoothness.** We calculate the mean and standard deviation of CDPAM along with the morphing path to validate smoothness, as $\text{CDPAM}_{mean \pm std} = \text{CDPAM}_{mean} \pm \text{CDPAM}_{std}$, where $\text{CDPAM}_{mean} = \frac{1}{N-1}\sum_{i=1}^{N-1} \text{CDPAM}(x^{(\alpha_i)}, x^{(\alpha_{i+1})})$, and $\text{CDPAM}_{std} = \sqrt{\frac{1}{N-1}\sum_{i=1}^{N-1}(\text{CDPAM}(x^{(\alpha_i)}, x^{(\alpha_{i+1})}) - \text{CDPAM}_{mean})^2}$. In timbre space study, we define *tim-*

*bral distance* by $\mathcal{L}_2^{timbre} = \frac{1}{N-1} \sum_{i=1}^{N-1} ||q^{(\alpha_{i+1})} - q^{(\alpha_i)}||_2$, where $q^{(\alpha_i)}$ represents the corresponding timber point of $x^{(\alpha_i)}$ in timbre space (McAdams et al., 1995). Lastly, to verify *reconstruction perceptual correspondence*, we denote $CDPAM_E$ that calculate CDPAM between $\{x^{(0)}, x^{(1)}\}$ and $\{\hat{x}^{(0)}, \hat{x}^{(1)}\}$, where $\hat{x}$ represents resynthesized end points when $\alpha = 0$ and $\alpha = 1$.

## 5.2 Timbral morphing for musical instruments

Sound morphing can allow timbral morphing between the sound of two known musical instruments, creating sounds from unknown parts of the timbre space (McAdams, 2013; McAdams & Goodchild, 2017). Timbral morphing for musical instruments involves transitioning between timbres of two different musical instruments to create a new sound. This new sound could possess characteristics of both original timbres as well as new timbral qualities between them, which usually applied to creative arts. In this experiment, we perform timbral morphing for isolated musical instruments given two recordings of the same musical composition played by different music instruments.

**Dataset.** To ensure high quality of paired composition musical instrument on timbral study, we selected 22 paired musical instrument composition samples from demonstration pages of musical timbre transform projects, MusicMagus (Zhang et al., 2024) and Timbrer (Kemppinen P., 2020). The paired samples have durations varies from 5s to 10s, with 16.0kHz and 44.01kHz, which involve 5 groups of instrument pairs: 2 paired samples of piano-violin; 10 paired samples of piano-guitar; 1 paired sample of taiko-hihat; 1 paired sample of piano-organ, and 8 paired samples of harp-kalimaba.

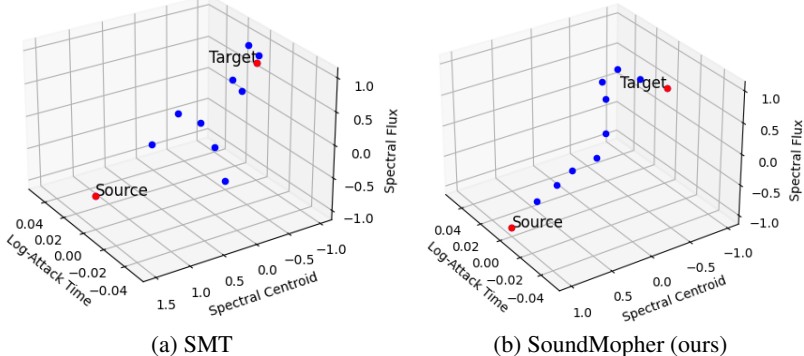

(a) SMT            (b) SoundMopher (ours)

Figure 1: Timbre space visualization of morph trajectories for piano-organ timbre morphing. Compared to SMT, SoundMorpher produces a smoother and continuous morph in the timbre space.

**Baseline.** We compare our method with Sound Morphing Toolbox (SMT) (Caetano, 2019), which is a set of Matlab functions targeting on musical instrument morphing that implement a sound morphing algorithm for isolated musical instrument sounds. Since SMT does not offer guidance for selecting perceptually uniform morph factors, we uniformly interpolate 11 morph factors in $[0, 1]$.

**Results and analysis.** We set $N = 11$ for SoundMorpher with an initial prompt '*a music composition by {instrument}*'. The comparison of our method and the baseline on timbral morphing is in Table 1. Overall, SoundMorpher demonstrates superior morphing quality compared to STM across various metrics, including audio quality, intermediateness, smoothness, and resynthesis quality [3], when applied to different types of musical instrument timbre morphing. Notably, STM fails in Taiko-Hihat timbral morphing due to significant high reconstruction perceptual error. In contrast, SoundMorpher maintains robustness across different types of musical instruments, making it a more flexible and efficient solution for timbral morphing applications on different types of musical instruments. Figure. 1 provides a visualization of normalized timbre space, illustrating morphing trajectories generated by SMT and SoundMorpher. The timbre space is defined by three important timbral features: Log-Attack Time, Spectral Centroid, and Spectral Flux (McAdams et al., 1995; McAdams, 2013). The SMT trajectory shows distinct steps, indicating that the transitions between each intermediate sound are relatively abrupt. The spacing between the blue points suggests that each step represents a significant change in timbre, which may result in a less smooth perceptual transition between two musical instruments. In contrast, the trajectory produced by SoundMopher demonstrates

---

[3]Since we perform timbral morphing within the same music composition, $MFCCs_\mathcal{E}$ may not a suitable metric under the same musical content. In contrast, we focus on evaluating smoothness and intermediateness.

Table 2: Environmental sound morphing with different types of environmental sounds.

| Category | FAD$_{category}$ | MFCCs$_{\mathcal{E}}\downarrow$ | FAD$\downarrow$ | FD$\downarrow$ | CDPAM$_T\downarrow$ | CDPAM$_{mean \pm std}\downarrow$ | CDPAM$_E\downarrow$ |
|---|---|---|---|---|---|---|---|
| Dog $\leftrightarrow$ Cat | 26.08 | 0.081 | 17.77 | 73.92 | 1.293 | $0.323 \pm 0.160$ | 0.236 |
| Laughing $\leftrightarrow$ Crying baby | 10.39 | 0.044 | 9.35 | 65.98 | 0.855 | $0.214 \pm 0.077$ | 0.289 |
| Church bells $\leftrightarrow$ Clock alarm | 68.29 | 0.058 | 22.89 | 75.77 | 2.205 | $0.551 \pm 0.299$ | 0.312 |
| Door knock $\leftrightarrow$ Clapping | 21.36 | 0.083 | 10.85 | 76.35 | 1.594 | $0.428 \pm 0.220$ | 0.321 |

a smoother curve. The points are more closely spaced, indicating more gradual changes between each intermediate timbre. This suggests that SoundMopher achieves a more continuous and natural-sounding morphing process, with each step being a smaller, more refined adjustment compared to SMT. Figure 7 and Figure 8 in the appendix provides additional visualization for this experiment.

### 5.3 ENVIRONMENTAL SOUND MORPHING

Environmental sounds are used in video game production to provide a sense of presence within a scene. For example, in video, AR and VR games, sound morphing could enhance user immersion by adapting audio cues to specific visual and interactive contexts. This means that it could be useful to morph between sonic locations, e.g., a city and a park, or between sound effects, e.g., different animal sounds to represent fantasy creatures. In this experiment, we perform cyclostationary morphing with $N = 5$ by SoundMorpher across various types of environmental sounds.

**Dataset.** We use ESC50 (Piczak, 2015) which consists of 5-second recordings organized into 50 semantic classes which loosely arranged into 5 major categories. We randomly select 4 major categories of scenarios to verify our method, including (1) Dog-Cat (animals voices), (2) Laughing-Crying baby (human sounds), (3) Church bells-Clock alarm (urban noise-interior sound), (4) Door knock-clapping (interior sounds-human sounds). Each category of scenarios contains 25 randomly selected audio pairs, thereby, 100 randomly paired samples in total.

**Results and analysis.** In this experiment, we use initial text prompt as '*a sound clip of {sound class}*'. Table 2 presents the results of applying SoundMorpher to various categories of environmental sounds. To quantify the semantic gap between sound scence classes, we calculate FAD between them as FAD$_{category}$. The results demonstrate SoundMorpher is capable of effectively morphing a wide range of environmental sounds. However, environmental sounds with a large semantic gap between categories can negatively impact the morphing quality. Additionally, we observe that the quantitative metrics for morphing quality and reconstruction perceptual errors in this experiment are higher than those for the timbre morphing task. One reason is the inherent complexity of environmental sounds, which often involve intricate physical events with significant temporal structure differences and background noises, making them more challenging to morph compared to musical data. Figure 9 in appendix provides spectrogram visualizations on environmental sound morphing.

### 5.4 MUSIC MORPHING

Film or game post-production often requires blending or fading between music tracks to seamlessly transition background music in between scenes. Music morphing transitions between two music compositions without cross fading, that is, each moment of the morphed music would be a single composition with elements that are perceptually in between both source and target music, rather than simply blending the two together. Different from timbral morphing, music morphing could ideally be accomplished with compositions from different genres and mixed musical instruments. In this experiment, we use SoundMorpher to perform dynamic morphing on music with $N = 15$.

**Dataset.** In this experiment, we randomly selected 50 sample pairs from 20 musical samples available on AudioLDM2 (Liu et al., 2024) demonstration page. These 10-second music compositions that span different genres and feature both single or mixed musical instrument arrangements.

**Results and analysis.** In this experiment, we select an initial text prompt as '*a sound clip of music composition*' to perform dynamic morphing. Even though this experiment contains morph complex music compositions with different music genres and music instruments, Table 3 shows our method still superiors on perceptual smoothly transiting source music to the target music and ensures correspondence, intermediateness and smoothness. Figure 2 provides strong visual evidence that the dynamic morphing method effectively transitions from the source to the target music while maintaining perceptual smoothness, correspondence, and intermediate transformations. The spectrogram

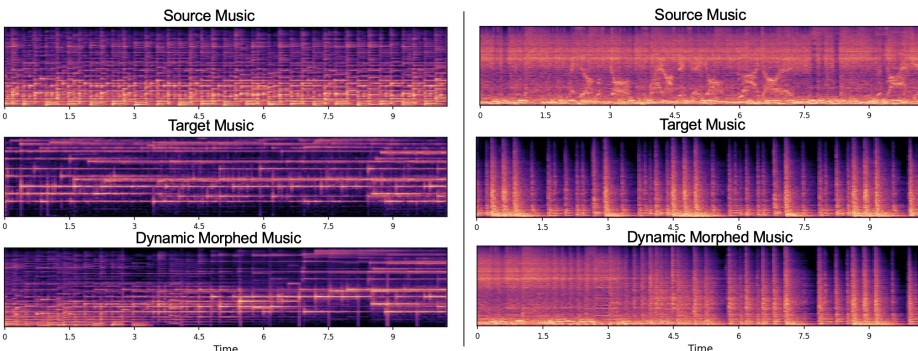

Figure 2: Visualization of dynamic morphed music with $N = 15$, source music and target music.

Table 3: Music morphing experimental results & ablation study for sound perceptual features, where N represents the number of components of PCA for reducing dimension of Mel-spectrogram.

| Feature | MFCCs$_{\mathcal{E}}$ ↓ | FAD ↓ | FD ↓ | CDPAM$_T$ ↓ | CDPAM$_{mean \pm std}$ ↓ | CDPAM$_E$ ↓ |
|---------|---------|-------|------|---------|---------|---------|
| Reduced Mel-Spec. (N=2) | 0.187 | 10.31 | 58.57 | 0.793 | $0.056 \pm 0.075$ | 0.182 |
| Reduced Mel-Spec. (N=3) | 0.151 | 10.76 | 59.39 | **0.779** | $\mathbf{0.055} \pm 0.079$ | 0.151 |
| Mel-Spec. | 0.056 | **9.85** | **56.09** | 0.847 | $0.068 \pm \mathbf{0.045}$ | 0.178 |
| MFCCs | **0.053** | 10.11 | 57.38 | 0.987 | $0.071 \pm 0.050$ | 0.156 |
| Spectral contras | 0.066 | 10.54 | 58.44 | 0.863 | $0.061 \pm 0.071$ | 0.155 |

illustrates that the morphed music transitions gradually, maintaining smooth spectral changes over time, which suggests the method successfully morphs the source into the target music.

Table 4: Mean opinion score study on environmental sound morphing and music morphing task.

| Task | Correspondence↑ | Intermediateness↑ | Smoothness↑ | Overall ↑ |
|------|---------|---------|---------|---------|
| Environmental sound morphing | $3.78 \pm 0.31$ | $3.67 \pm 0.40$ | $3.57 \pm 0.47$ | $3.67 \pm 0.39$ |
| Music morphing | $3.81 \pm 0.39$ | $3.55 \pm 0.51$ | $3.49 \pm 0.48$ | $3.62 \pm 0.46$ |

## 5.5 DISCUSSION

**Mean opinion score (MOS) study.** We conducted a MOS study as a subjective evaluation for the correspondence, intermediateness, and smoothness of morphed results from SoundMorpher. The study involved 21 volunteers, and detailed methodology is in Appendix 9. As shown in Table 4, the results suggest that SoundMorpher is versatile, performing similarly well across both music and environmental sound morphing tasks, with no significant differences observed in the overall MOS. This consistency in scores indicates that SoundMorpher effectively handles the unique challenges posed by the distinct characteristics of music and environmental sounds, such as the continuous nature of music compared to the more segmented structure of environmental sounds. Despite the objective metric results showing clear differences between the two tasks, the human evaluation suggests that SoundMorpher remains robust across different sound types. One possible interpretation is that our objective metrics are more sensitive to variations in the measured aspects than participants.

**Model comparison.** We compare SoundMorpher with a concurrent work, MorphFader (Kamath et al., 2024), based on the criteria outlined in Section 3.1. MorphFader relies on a pre-trained AudioLDM (Liu et al., 2023) and perform

Table 5: Comparison with MorphFader

| Method | CDPAM$_T$ | CDPAM$_{mean \pm std}$ | MFCCs$_{\mathcal{E}}$ |
|--------|---------|---------|---------|
| MorphFader | 0.972 | $0.243 \pm 0.139$ | 0.065 |
| SoundMorpher | 0.935 | $0.226 \pm 0.162$ | 0.065 |

morphing by text instructions. We compare with 7 examples provided on the demonstration of MorphFader, that MorphFader uniformly samples 5 morph factors in $[0, 1]$, resulting in a morph path with $[0, 0.25, 0.5, 0.75, 1]$. In contrast, SoundMorpher finds $\alpha$ values according to constant $\Delta p$ with 5 uniformly interpolate $p$ points by binary search. As in Table 5, SoundMorpher produces a smoother and perceptual intermediates morphing than MorphFader. See Appendix 10 for details.

**Ablation study on sound perceptual features.** We verified the perceptual feature in SPDP in music morphing task. We select alternative music information retrieval features (MIR) including MFCCs with 13 coefficients (Logan et al., 2000), and spectral contrast (Jiang et al., 2002). We use principal component analysis (PCA) to reduce the dimensionality of Mel-spectrogram to further capture variation of spectral content over time, which is referred to as reduced Mel-Spec. (Stevens et al., 1937; Casey et al., 2008; Jiang et al., 2002). Table 3 shows performance comparisons of SoundMor-

Table 6: Experiment and ablation study results on music morphing.

| Init. text | T = 20 | T =100 | MFCCs$_\mathcal{E}$ ↓ | FAD ↓ | FD ↓ | CDPAM$_T$ ↓ | CDPAM$_{mean\pm std}$ ↓ | CDPAM$_E$ ↓ |
|---|---|---|---|---|---|---|---|---|
| Informative | ✓ | | 0.047 | 10.21 | 56.62 | 1.213 | 0.086 ± 0.069 | 0.166 |
| Informative | | ✓ | 0.044 | 10.21 | 56.13 | 1.077 | 0.084 ± 0.066 | 0.155 |
| Uninformative | ✓ | | 0.057 | 10.37 | 55.89 | 1.036 | 0.074 ± 0.049 | 0.211 |
| Uninformative | | ✓ | 0.056 | 9.85 | 56.09 | 0.847 | 0.068 ± 0.045 | 0.178 |

pher with different features, SoundMorpher with Mel-spectrogram achieves better morphing quality in terms of correspondence and smoothness variation with smaller FAD, FD and CDPAM$_{std}$. While Mel-spectrogram yields higher CDPAM$_{mean}$, CDPAM$_T$ and MFCCs$_\mathcal{E}$ compared to reduced Mel-Spec. and MFCCs, the differences in metric values are not significant. However, the overall morph quality with Mel-spectrogram is consistently better than other features. This suggests Mel-spectrogram, as a pseudo-3D representation, provide more perceptual and semantic information, which contributes to improve morph quality compared to higher-level features.

**Uninformative v.s. informative initial text prompt.** Complex audio usually cannot easily yield precise information to users. For example, it is a challenge for non-professional users to describe the genre of a music. We conduct an ablation study for initial text prompt on music morphing to verify effectiveness of text-guided conditional embedding optimization. We use a general initial text prompt, *'a sound clip of music composition.'*, as an uninformative initial prompt. And we use the given text prompts in AudioLDM2 [4] as informative inital prompts. As in Table 6, informative initial text prompts may help with resynthesis quality and further improves morph correspondence. Despite the improved resynthesis quality with informative initial text prompts, the results show a decline in morphing intermediateness and smoothness. One possible reason is the better resynthesis quality makes the resynthesis endpoints more distinct (i.e., larger semantic gap), which could lead to slight decline in intermediateness and smoothness. However, the performance difference on initial text prompts is not significant which illustrates effectiveness of conditional embedding optimization.

**Inference steps.** In our experiment, we follow the configuration of Zhang et al. (2024) and set DDIM steps to 100. To verify whether DDIM steps affect SoundMorpher performances, we compare with 20 DDIM steps in Table 6. Larger inference step seems to help for reconstruction quality and slightly imporves morph quality, however, performance differences between inference steps are not significant. This indicates SoundMorpher is robust for inference steps, and we extend this ablation study on environmental sound morphing task in Appendix 11.1. Thus, we suggest selecting a suitable DDIM step to trade-off overall binary search algorithm time-consuming and morph quality.

**Limitations.** The current implementation of SoundMorpher based on AudioLDM2 with 16.0kHz sampling rate, which may limit output audio quality. The conditional embeddings optimization only applies to sounds that can be produced by AudioLDM2. Sound examples that close to white noise, such as pure wind blowing used in MorphGAN (Gupta et al., 2023) are not easily generated by AudioLDM2, which makes the conditional embedding optimization produce low quality resynthesis sounds. We also observed that input sounds with a large semantic gap (e.g., Church bells-Clock alarm in Table 2) result in lower morphing quality. Furthermore, we observed when two audios exhibit significant temporal structure differences, such as environmental sounds, SoundMorpher may produce abrupt transitions, see Appendix 11.4 and Figure 6 for further details.

## 6 CONCLUSION

We propose SoundMorpher, a sound morphing method base on a pretrained diffusion model that produces perceptually uniform morphing trajectories. Unlike existing methods, we refined the sound morphing problem and explored an explicit connection between morph factor and perceptual stimuli of morphed results which offers better flexibility and higher morphing quality, making it adaptable to various morphing methods and real-world scenarios. We validate SoundMorpher by a series of objective quantitative metrics as well as mean opinion score study following criteria proposed by Caetano & Osaka (2012). These quantitative objective metrics may help to formalize future studies on sound morphing evaluation. Furthermore, we demonstrated that SoundMorpher can be applied to wide range of real-world applications in our experiments and conducted in-depth discussions. SoundMorpher also has the potential to achieve voice morphing, as its foundational model AudioLDM2 supports speech generation; however, we leave this exploration for future work.

---

[4]https://audioldm.github.io/audioldm2/

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

APPENDIX

# 7 SOUNDMORPER IMPLEMENTATION DETAILS

This section provides further details on SoundMorpher pipeline and implementations.

## 7.1 OVERALL PIPELINE OF SOUNDMORPHER

The overall pseudo pipeline for SoundMorpher is provided in Algorithm 1, where the overall pipeline of SoundMorpher contains three parts: (1) Conditional embedding optimization. (2) Model adaptation. (3) perceptually uniform binary search with constant SPDP increment.

---

**Algorithm 1** Pipeline of SoundMorpher

---

**Require:** A pre-trained AudioLDM2 pipeline including a pre-trained VAE with a encoder $g_\theta$ and a decoder $g_\phi$, a pre-trained latent diffusion model $\epsilon_\theta$, a pre-trained T5 model $f_\phi$, and a pre-trained GPT-2 model $f_\varphi$. Learning rates $\eta_1, \eta_2$. Source and target audios $x^{(0)}$ and $x^{(1)}$. An initial text prompts $y$. perceptually uniform interpolation number $N$. Tolerance error for binary search $\epsilon_{tolerance}$. Number of training steps for text inversion for conditional embedding optimization $T_{inv}$. Number of training steps for model adaptation $T_{adapt}$. LoRA rank $r$, Number of steps for DDIM $T$.

**Ensure:** Start morph factor $\alpha_{start} = 0$, end morph factor $\alpha_{end} = 1$. Start perceptual point $p_{start} = [0, 1]$, and end perceptual point $p_{end} = [1, 0]$.

**Initialize:** $z_0^{(0)} = g_\theta(x_0^{(0)})$, $z_0^{(1)} = g_\theta(x_0^{(1)})$; $E^0 = [f_\phi(y), f_\varphi(y)]$, $E^1 = [f_\phi(y), f_\varphi(y)]$;

  # Step 1: Text-guided conditional embedding optimization.

  **for** $i$ **from** 1 **to** $T_{inv}$ **do**

    Randomly sample time step $t$ and random noise $\epsilon_t \sim N(0, I)$.

    Adding noise to data $z_t^{(0)} \leftarrow \sqrt{\gamma_t} z_0^{(0)} + \sqrt{(1-\gamma_t)}\epsilon_t$, $z_t^{(1)} \leftarrow \sqrt{\gamma_t} z_0^{(1)} + \sqrt{(1-\gamma_t)}\epsilon_t$.

    $E^{(0)} \leftarrow E^{(0)} - \eta_1 \nabla_{E^{(0)}} \mathcal{L}_{DPM}(z_0^{(0)}, E^{(0)}; \theta)$.

    $E^{(1)} \leftarrow E^{(1)} - \eta_1 \nabla_{E^{(1)}} \mathcal{L}_{DPM}(z_0^{(1)}, E^{(1)}; \theta)$.

  **end for**

  # Step 2: Model adaptation with LoRA.

  **for** $i$ **from** 1 **to** $T_{adapt}$ **do**

    Model adaptation with LoRA according to Equation 9 and Equation 10 with $\eta_2$ learning rate.

  **end for**

  # Step 3: Perceptual-uniform binary search with constant SPDP increment.

  Obtaining initial latent states $z_T^{(0)}$ and $z_T^{(1)}$ by Equation 4

  $p_{list} \leftarrow \text{ConstantSPDP}(N, p_{start}, p_{end})$       ▷ Obtain target SPDP points by Algorithm 2

  $\alpha_{list} \leftarrow \text{BinarySeach}(\alpha_{start}, \alpha_{end}, p_{list}, \epsilon_{tolerance})$

  **for** $\alpha$ **in** $\alpha_{list}$ **do**

    $E^{(\alpha)} \leftarrow (1-\alpha)E^{(0)} + \alpha E^{(1)}$

    $z_T^{(\alpha)} \leftarrow \frac{sin(1-\alpha)w}{sinw} z_T^{(0)} + \frac{sin\alpha w}{sinw} z_T^{(1)}$

    **for** $t$ **from** T **to** 1 **do**

      $z_{t-1}^{(\alpha)} \leftarrow \sqrt{\gamma_{t-1}}(\frac{z_t - \sqrt{1-\gamma_t}\hat{\epsilon}_\theta^{(t)}(z_t, E^{(\alpha)})}{\sqrt{\gamma_t}}) + \sqrt{1-\gamma_{t-1}}\hat{\epsilon}_\theta^{(t)}(z_t, E^{(\alpha)})$

    **end for**

  **end for**

  $x^{(\alpha)} \leftarrow \text{vocoder}(g_\phi(z_0^{(\alpha)}))$     ▷ Decode latent variable and obtain audio waveform by a vocoder.

  **return** $\{x^{(\alpha)}\}_{\alpha \in \alpha_{list}}$

---

## 7.2 MODEL ADAPTATION WITH LORA

In the task of image morphing, Yang et al. (2023) indicate adapting the model to the input pair helps to limit the degree of morphed variation by suppressing high-density regions that are not related to

the given images. Compared to vanilla fine-tuning approaches, LoRA has advantages in training efficiency with injecting limited trainable parameters. The model adaptation can be defined as

$$\min_{\Delta\theta'}\mathcal{L}_{DPM}(z_0^{(0)}, E^{(0)}; \theta + \Delta\theta') + \min_{\Delta\theta'}\mathcal{L}_{DPM}(z_0^{(1)}, E^{(1)}; \theta + \Delta\theta') \text{ s.t. } \text{rank}(\Delta\theta') = r \quad (9)$$

where $r$ represents LoRA rank.

**Unconditional Bias Correction.** To achieve high text alignment during inference time, we use additional LoRA parameters $\Delta\theta_0$ with a small rank $r_0$ to perform bias correction, as

$$\min_{\Delta\theta_0}\mathcal{L}_{DPM}(z_0^{(0)}, \varnothing; \theta + \Delta\theta_0) + \min_{\Delta\theta_0}\mathcal{L}_{DPM}(z_0^{(1)}, \varnothing; \theta + \Delta\theta_0) \text{ s.t. } \text{rank}(\Delta\theta_0) = r_0 \quad (10)$$

During inference, with $\theta' = \theta + \Delta\theta'$ and $\theta_0 = \theta + \Delta\theta_0$, the noise prediction becomes

$$\hat{\epsilon}_\theta(z_t, t, E) := w\epsilon_{\theta'}(z_t, t, E) + (1-w)\epsilon_{\theta_0}(z_t, t, \varnothing). \quad (11)$$

In our experiment, we set $r = 4$ and $r_0 = 2$. Although Yang et al. (2023) provide a heuristic suggestion for setting the LoRA rank value for image morphing task, however, we further investigate the relationship between LoRA rank $r$ and method performance in Table 8 in sound morphing task and discussion in Appendix 11.2.

### 7.3 PERCEPTUALLY UNIFORM BINARY SEARCH WITH CONSTANT SPDP INCREMENT

Algorithm 2 provides detail pesudocodes for how to implement perceptually uniform binary search with constant SPDP increment. This pseudo algorithm includes two steps, firstly, compute constant SPDP increment according to interpolte point number $N$ and obtain $N$ target SPDP points as $\{p_i\}_{i=1}^N$. Secondly, perform binary search to find correponding morph factor $\alpha$ series $\{\alpha_i\}_{i=1}^N$ according to $\{p_i\}_{i=1}^N$.

### 7.4 SOUND MORPHING WITH DISCRETE $\alpha$ SERIES

This section provide detail pseudo algorithm to perform different types of sound morphing methods:

1. Static morphing: see Algorithm 3;
2. Cyclostationary morphing: see Algorithm 4;
3. Dynamic morphing: see Algorithm 5;

However, SoundMorpher is not restricted to the aforementioned sound morphing methods; it can be extended to other approaches, such as warped dynamic morphing, by concatenating the original dynamic morphing result with its reversed counterpart. We leave this exploration for future work.

### 7.5 CONVEX CFG SCHEDULING FOR QUALITY BOOSTING

**Background for Classifier-free Guidance (CFG).** Controllable generation can be achieved by using guidance at each sampling step in diffusion model. When a conditional and unconditional diffusion models are jointly trained, samples can be obtained by CFG (Ho & Salimans, 2022). In AudioLDM2 (Liu et al., 2024), the the conditional and unconditional noise esitimation becomes

$$\hat{\epsilon}_\theta(z_t, t, E) := w\epsilon_\theta(z_t, t, E) + (1-w)\epsilon_\theta(z_t, t, \varnothing) \quad (12)$$

where $w$ determines the guidance scale.

**Convex CFG scheduling.** Following Yang et al. (2023), we involve a convex CFG scheduling in SoundMorpher to boost morphing quality which is defined as

$$w_\alpha = w_{max} - 2(w_{max} - w_{min})|\alpha - 0.5| \quad (13)$$

where $w$ is the guidance scale, $\alpha$ is the morph factor. $w_{max}$ and $w_{min}$ are predefined maximum and minimum guidance scales. We discussed the impact of guidance scales in SoundMorpher in Appendix 11.3.

---

**Algorithm 2** Pseudo algorithm for perceptually uniform binary search with constant SPDP increment

---

**Require:** $\alpha_{start}$: starting alpha value; $\alpha_{end}$: ending alpha value; $N$: number of interpolations; source audio $x^{(0)}$; target audio $x^{(1)}$;

**Ensure:** $p_{list} = []$; $p_{start} = [0, 1]^T$; $p_{end} = [1, 0]^T$; $\alpha_{list} = []$;
   # Step 1: Obtain target SPDP points with constant increment.
   **Procedure** ConstantSPDP($N, p_{start}, p_{end}$)
   **for** $i$ **from** 1 **to** $N - 1$ **do**                 ▷ Find target SPDP points with constant $\Delta p$.
        $t \leftarrow \frac{i}{N-1}$;
        $p_i \leftarrow (1 - t) \times p_{start} + t \times p_{end}$
        $p_{list} \leftarrow p_{list} \cup [p_i]$;
   **end for**
   # Step 2: Perform binary search given target SPDP points with constant $\Delta p$.
   **Procedure** BinarySearch($\alpha_{start}, \alpha_{end}, x^{(0)}, x^{(1)}, p_{list}, \epsilon_{tolerance}$);
   $\alpha_{list} \leftarrow [\alpha_{start}]$;
   $\alpha_{cur} \leftarrow \alpha_{start}$;
   **for** $p_i$ **from** $p_1$ **to** $p_{N-2}$ **do**
        $p_{target} \leftarrow p_i$
        $\alpha_{t1} \leftarrow \alpha_{cur}$;
        $\alpha_{t2} \leftarrow \alpha_{end}$;
        $\alpha_{mid} \leftarrow \frac{\alpha_{t1} + \alpha_{t2}}{2}$;
        $p_{mid} \leftarrow SPDP(x^{\alpha_{mid}}, x^{(0)}, x^{(1)})$              ▷ Compute SPDP by Equation 7.
        **while** $|p_{mid} - p_i| > \epsilon_{tolerance}$ **do**                 ▷ Perform binary search
           **if** $p_{mid} > p_{target}$ **then**
              $\alpha_{t2} \leftarrow \frac{\alpha_{t1} + \alpha_{t2}}{2}$
           **else**
              $\alpha_{t1} \leftarrow \frac{\alpha_{t1} + \alpha_{t2}}{2}$
           **end if**
           $\alpha_{mid} \leftarrow \frac{\alpha_{t1} + \alpha_{t2}}{2}$
           $p_{mid} \leftarrow SPDP(x^{\alpha_{mid}}, x^{(0)}, x^{(1)})$
        **end while**
        $\alpha_{list} \leftarrow \alpha_{list} \cup [\alpha_{mid}]$              ▷ Append the reult to $\alpha_{list}$.
   **end forreturn** $\alpha_{list}$;

---

**Algorithm 3** Pseudo algorithm for static morphing

---

**Require:** Source audio $x^{(0)}$, target audio $x^{(1)}$, specific SPDP point $p$, tolerance error for binary search $\epsilon_{tol}$, number of steps for DDIM $T$, VAE decoder $g_\phi$.

**Ensure:** $\alpha_{start} = 0, \alpha_{end} = 1$;
   $\alpha \leftarrow$ BinarySearch($\alpha_{start}, \alpha_{end}, x^{(0)}, x^{(1)}, p, \epsilon_{tolerance}$);
   $E^{(\alpha)} \leftarrow (1 - \alpha)E^{(0)} + \alpha E^{(1)}$;
   $z_T^{(\alpha)} \leftarrow \frac{sin(1-\alpha)w}{sinw} z_T^{(0)} + \frac{sin\alpha w}{sinw} z_T^{(1)}$;
   **for** $t$ **from** T **to** 1 **do**
        $z_{t-1}^{(\alpha)} \leftarrow \sqrt{\gamma_{t-1}}(\frac{z_t - \sqrt{1-\gamma_t}\hat{\epsilon}_\theta^{(t)}(z_t, E^{(\alpha)})}{\sqrt{\gamma_t}}) + \sqrt{1 - \gamma_{t-1}}\hat{\epsilon}_\theta^{(t)}(z_t, E^{(\alpha)})$;
   **end for**
        **return** $x^{(\alpha)} \leftarrow$ vocoder($g_\phi(z_0^{(\alpha)})$);

---

**Algorithm 4** Pseudo algorithm for cyclostationary morphing

**Require:** Source audio $x^{(0)}$, target audio $x^{(1)}$, number of interpolations $N$, tolerance error for binary search $\epsilon_{tol}$, number of steps for DDIM $T$, VAE decoder $g_\phi$.
**Ensure:** $\alpha_{start} = 0, \alpha_{end} = 1, p_{start} = [0, 1], p_{end} = [1, 0], x_{list} = [];$
    $p_{list} \leftarrow \text{ConstantSPDP}(N, p_{start}, p_{end});$
    **for** $p_i$ **in** $p_{list}$ **do**
        $\alpha_i \leftarrow \text{BinarySearch}(\alpha_{start}, \alpha_{end}, x^{(0)}, x^{(1)}, p_i, \epsilon_{tolerance});$
    **end for**
    **for** $\alpha_i$ **in** $\alpha_{list}$ **do**
        $E^{(\alpha_i)} \leftarrow (1 - \alpha_i)E^{(0)} + \alpha_i E^{(1)};$
        $z_T^{(\alpha_i)} \leftarrow \frac{sin(1-\alpha_i)w}{sinw} z_T^{(0)} + \frac{sin\alpha_i w}{sinw} z_T^{(1)};$
        **for** $t$ **from** T **to** 1 **do**
            $z_{t-1}^{(\alpha)} \leftarrow \sqrt{\gamma_{t-1}}(\frac{z_t - \sqrt{1-\gamma_t}\hat{\epsilon}_\theta^{(t)}(z_t, E^{(\alpha)})}{\sqrt{\gamma_t}}) + \sqrt{1-\gamma_{t-1}}\hat{\epsilon}_\theta^{(t)}(z_t, E^{(\alpha)});$
        **end for**
        $x^{(\alpha_i)} \leftarrow \text{vocoder}(g_\phi(z_0^{(\alpha_i)}));$
        $x_{list} \leftarrow x_{list} \cup x^{(\alpha_i)}$
    **end for**
        **return** $x_{list}$

**Algorithm 5** Pseudo algorithm for dynamic morphing

**Require:** A list of cyclostationary morphed results $x_{list}$, number of interpolation points $N$, audio length $T_{audio}$.
**Ensure:** Dynamic morphing output $x_{dy} = []$
    **for** $i$ **from** 0 **to** N-1 **do**
        $L_{clip} = T_{audio}//N$
        $x_{seg} = x_i[i \times L_{clip} : (i+1) \times L_{clip}];$
        $x_{dy} \leftarrow \text{concat}(x_{dy}, x_{seg})$
    **end for**
        **return** $x_{dy}$

# 8 EXPERIMENT SETUP AND IMPLEMENTATION DETAILS

## 8.1 EXPERIMENT SETUP

We perform our experiment on one NVIDIA GeForce RTX 3090 with GPU 24GB memory. Following configuration in Yang et al. (2023), we use AdamW optimizer (Loshchilov & Hutter, 2017) with learning rate 0.002 and 2500 steps to perform conditional embedding optimization. We use LoRA (Hu et al., 2021) with $r = 4$ and $r_0 = 2$ to perform model adaptation, the LoRA is trained by Adam optimizer with 0.001 learning rate. We trained 150 steps for the LoRA injected trainable paramters for model adaptation and 15 steps for LoRA injected trainable parameters for unconditional bias correction. For convex CFG scheduling, we set $w_{max} = 3.5$ and $w_{min} = 1.5$ for timbral morphing and environmental sound morphing task, and $w_{max} = 4$ and $w_{min} = 1.5$ for music morphing task.

## 8.2 IMPLEMENTATION DETAILS FOR MFCCs$_\mathcal{E}$ CALCULATION

The goal of designing MFCCs$_\mathcal{E}$ feature is to verify the *correspondence* of morphed samples as an objective metric. Let $N$ be an odd integer, We define a series of perceptually uniform morphing results $\{x^{(\alpha_i)}\}_{i=1}^N$ by the proposed method with source and target audio $x^{(0)}$ and $x^{(1)}$, where $i$ in the range of 1 to $N$ and the source audio $x^{(0)}$ and the target audio $x^{(1)}$ has different contents. Each element $x^{(\alpha_i)}$ in the series represents a morphed result corresponding to a specific morphing parameter $\alpha_i$, thus the set can be written as

$$\{x^{(\alpha_i)}\}_{i=1}^N = \{x^{(\alpha_1)}, x^{(\alpha_2)}, ..., x^{(\alpha_N)}\} \tag{14}$$

The MFCCs$_\mathcal{E}$ is computed by

$$\text{MFCCs}_\mathcal{E} = \text{abs}(\frac{||m^{(\frac{N+1}{2})} - m^{(0)}||_2}{||m^{(\frac{N+1}{2})} - m^{(0)}||_2 + ||m^{(\frac{N+1}{2})} - m^{(1)}||_2} - 0.5) \tag{15}$$

where $m^{(i)}$ represents the extracted MFCCs feature of the $i^{th}$ morphed results in the series $x^{(\alpha_i)}$, $m^{(0)}$ and $m^{(1)}$ represents MFCCs feature of $x^{(0)}$ and $x^{(1)}$. This metric aims to evaluate spectrogram content of the perceptual midpoint result $x^{(\frac{N+1}{2})}$ between two end points $x^{(0)}$ and $x^{(1)}$. Ideally, we wish the midpoint morphed result contains half-and-half content on two end points. The larger MFCCs$_{error}$ indicates the content consistency is far away than the midpoint (i.e., 0.5). We extract MFCCs feature with 13 coefficients to compute MFCCs$_\mathcal{E}$.

## 8.3 IMPLEMENTATION DETAILS FOR COMPUTING FAD AND FD

FAD and FD are commonly used in audio generation tasks to measure the quality of synthesized audio. Given two branches of audio groups, synthesized audios and real audios, these metrics offer a comprehensive assessment of the global quality by evaluating how closely the synthesized audio matches the distribution of real audio.

In our experiment, we calculate FAD and FD between morphed audios $\{x^{(\alpha_i)}\}_{i=1}^N$ and sourced audios to reflect correspondence of morphing and audio quality of morphed results. Smaller the FAD and FD values indicate the morphed audios has smaller semantic distribution gap between real sourced audios, suggesting that the morphed audios are more natural and exhibit semantic consistency.

Specifically, in the timbral morphing task, we categorize source audios based on groups of musical instruments, such as piano-guitar, harp-kalimba, etc. We then compute FAD and FD values between a consentive morphed path $\{x^{(\alpha_i)}\}_{i=1}^N$ and the corresponding instrument group to which the endpoint audios belong. Similarity, in the case of environmental sound morphing task, we classify source audios according to sound scene categories and compute FAD and FD values between a consentive morphed path $\{x^{(\alpha_i)}\}_{i=1}^N$ and the corresponding sounds to which the endpoint audios belong to the categories. And in music morphing task, we calculate FAD and FD values between morphed audios $\{x^{(\alpha_i)}\}_{i=1}^N$ with 20 samples of sourced music.

### 8.4 IMPLEMENTATION DETAILS FOR TIMBRAL SPACE CALCULATION

In timbral morphing task, we calculate timbral distance as an additional metric for evaluating smoothness of morphing. Refering to McAdams (2013); McAdams et al. (1995), we compute log attack time, spectral centroid and spectral flux from audio signal as the first, second and third dimension for plotting the timbre space. To plot the timbre space as in Figure 1, we normalized the value of each point $q$ into values between [-1,1].

## 9 FURTHER INFORMATION OF MEAN OPINION SOCRE STUDY

This section provide further implementation deatils for mean opinion score study. Therefore, we designed a 30 minues survey with 30 groups of evaluation questions. We randomly select 16 groups of cyclostationary morphing results (4 samples for each category) from the task of environmental sound morphing, and 14 groups of dynamic morphing results from the task music morphing, resulting in 30 groups of morphed examples in total. Each group has around 30 seconds audio durations, thereby, the quesionnaire takes around 30 minutes to complete (including time for reading the participants information sheet and response the questions).

For each group, we designed three questions for participants to give their opinion score regarding to correspondence, intermediateness, and smoothness. Details are

1. Content Consistency: How consistent is the content of the morphed sound with the content of both the source and target sounds?

2. Perceptual Consistency: How much does the morphed sound seem to be in between the source and target sounds?

3. Smoothness of Transition: How smoothly does the transition occur in the morphed sound from the source sound to target sound?

Participants give score according to following scale:

- 1 - Bad
- 2 - Poor
- 3 - Fair
- 4 - Good
- 5 - Excellent

Figure 3 and Figure 4 provide example questions in case of environmental sound morphing sample and music dynamic morphing sample in our MOS study.

We distributed our questionnaire link to some facebook groups and collected 21 completed responses from volunteer participants in this MOS study. During this study, only the opinion score participants provided are collected, we didn't collect any participants' personal information.

## 10 FURTHER INFORMATION OF MODEL COMPARISON WITH MORPHFADER

### 10.1 EXPERIMENTAL DETAILS

Due to MorphFader hasn't open source their method, we make comparison with 7 pairs of morphing examples on its demonstration page [5]. To have a fair comparison, We set DDIM inversion step as $T = 100$ and match number of interpolation $N = 5$ as in MorphFader. Due to we don't have original source and target audios for their demonstrations, we cannot compute $\text{CDPAM}_E$, FAD and FID in

---

[5]https://pkamath2.github.io/audio-morphing-with-text/webpage/audio-morphing.html

Please listen to the following audios by order and answer the following questions

**1. Source sound**

▶ 0:00 / 0:05 ──────── 🔊 ⋮

**2. Morphed sound**

▶ 0:00 / 0:05 ──────── 🔊 ⋮

**3. Morphed sound**

▶ 0:00 / 0:05 ──────── 🔊 ⋮

**4. Morphed sound**

▶ 0:00 / 0:05 ──────── 🔊 ⋮

**5. Target sound**

▶ 0:00 / 0:05 ──────── 🔊 ⋮

Question 1: How consistent is the content of the morphed sounds with the content of both the source and target sounds?

| Bad | Poor | Fair | Good | Excellent |
|-----|------|------|------|-----------|
| ○ | ○ | ○ | ○ | ○ |

Question 2: How much does the morphed sound seem to be in between the source and target sounds?

| Bad | Poor | Fair | Good | Excellent |
|-----|------|------|------|-----------|
| ○ | ○ | ○ | ○ | ○ |

Question 3: How smoothly does the transition occur in the morphed sounds from the source audio to target sounds?

| Bad | Poor | Fair | Good | Excellent |
|-----|------|------|------|-----------|
| ○ | ○ | ○ | ○ | ○ |

Figure 3: Example questions for a group of environmental sound morphing sample in MOS study.

Please listen to the following source and target sounds at first

**Source sound**

▶ 0:00 / 0:10 ——————— 🔊 ⋮

**Target sound**

▶ 0:00 / 0:10 ——————— 🔊 ⋮

The morphed sound gradually transits from source sounds to target sounds over time.

Please listen to the morphed sound and answer the following questions

**Morphed sound**

▶ 0:00 / 0:10 ——————— 🔊 ⋮

Question 1: How consistent is the content of the morphed sound with the content of both the source and target sounds?

| Bad | Poor | Fair | Good | Excellent |
|-----|------|------|------|-----------|
| ○ | ○ | ○ | ○ | ○ |

Question 2: How much does the morphed sound seem to be in between the source and target sounds?

| Bad | Poor | Fair | Good | Excellent |
|-----|------|------|------|-----------|
| ○ | ○ | ○ | ○ | ○ |

Question 3: How smoothly does the transition occur in the morphed sound from the source sound to target sound?

| Bad | Poor | Fair | Good | Excellent |
|-----|------|------|------|-----------|
| ○ | ○ | ○ | ○ | ○ |

Figure 4: Example questions for a group of music dynamic morphing sample in MOS study.

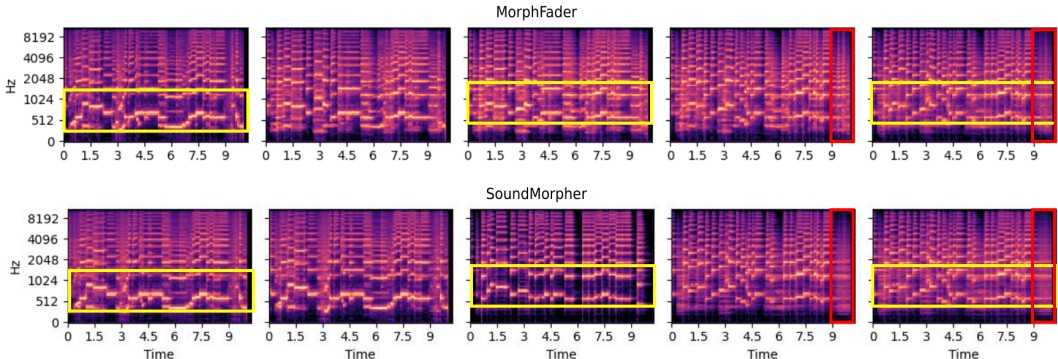

Figure 5: Visualization of spectrogram for morphed results compred with MorphFader.

this comparison. In this model comparison, we aim to validate how assumption that there exists a non-linear relationship between morph factor $\alpha$ and perceptual stimuli of the morphed result $p$. Therefore, simply uniformly increasing morph factor $\alpha$ cannot achieve truely perceptually smooth transitions. Fruthermore, this comparison showcases SoundMorpher's superior on producing sound morphing sequences with constant perceptual stimuli increment, which cannot achieves by simply controlling morph factors.

In this experiment, we use demonstrated audios with $\alpha = 0.0$ and $\alpha = 1.0$ as our input target and source audio. We set the source and target text prompt provided in MorphFader demonstration page as the initial text prompt for LOA feature optimization. We set $scale_{max} = 4$ and $scale_{min} = 1.5$.

## 10.2 Further anaylsis

Figure 5 provides a visualization comparison of a pair of audio samples between MorphFader and SoundMorpher. We analysis Figure 5 according to two aspects: correspondence and smoothness.

**Frequency Band Stability.** Yellow rectangles in Figure 5 highlight frequency band intensity cross morphing results of MorphFader and SoundMorpher. The intensity of the frequency bands within the yellow rectangle changes more abruptly for MorphFader, which could suggest that the morphing process introduces inconsistencies in the spectral content. In contrast, yellow rectangles in Sound-Morpher are more stable and consistent across time. The transitions between different frequency bands appear smoother, with fewer abrupt changes in intensity. This suggests that SoundMorpher maintains better spectral consistency during the morphing process, with smoother transitions between different timbral characteristics.

**Transition Smoothness.** As red rectangles indicate, MorphFader introduces more abrupt changes at the end of the morphing sequence. The spectral lines do not gradually transition; instead, there is a noticeable shift in the pattern, indicating less smoothness on transition. In contrast, SoundMorpher shows a more gradual and consistent transition within the red rectangles. The spectral patterns remain more stable and exhibit smoother transitions towards the end of the morphing sequence.

Overall, SoundMorpher appears to provide a more seamless and stable morphing process. The transitions are smoother, and the spectral content is more consistent across the morphing stages.

## 11 Further discussion

### 11.1 Ablation study on inference steps for environmental sound morphing task

This section provides a supplementary ablation study on the inference steps for the environmental sound morphing task. Table 7 presents the results of this ablation. Similar to the findings in the music morphing task, we observe no significant performance difference between using larger and

Table 7: Ablation study on inference steps for environmental sound morphing with different types of environmental sounds.

| | T= 20 | T= 100 | FAD↓ | FD↓ | CDPAM$_T$↓ | CDPAM$_{mean \pm std}$ ↓ | MFCCs$_\mathcal{E}$↓ | CDPAM$_E$↓ |
|---|---|---|---|---|---|---|---|---|
| Dog ↔ Cat | | ✓ | 17.77 | 73.92 | 1.293 | 0.323 ± 0.160 | 0.081 | 0.236 |
| | ✓ | | 18.27 | 81.60 | 1.172 | 0.293 ± 0.149 | 0.052 | 0.242 |
| Laughing ↔ Crying baby | | ✓ | 9.35 | 65.98 | 0.855 | 0.214 ± 0.077 | 0.044 | 0.289 |
| | ✓ | | 7.82 | 68.17 | 0.832 | 0.208 ± 0.115 | 0.078 | 0.250 |
| Church bells ↔ Clock alarm | | ✓ | 22.89 | 75.77 | 2.205 | 0.551 ± 0.299 | 0.058 | 0.312 |
| | ✓ | | 25.23 | 77.84 | 2.205 | 0.551 ± 0.304 | 0.056 | 0.352 |
| Door knock ↔ Clapping | | ✓ | 10.85 | 76.35 | 1.594 | 0.428 ± 0.220 | 0.083 | 0.321 |
| | ✓ | | 13.11 | 80.58 | 1.734 | 0.433 ± 0.281 | 0.118 | 0.281 |

Table 8: Ablation study on model adaptation with different LoRA rank $r$ on music morphing. Rank with $-$ represents results without LoRA model adaptation.

| Rank | MFCCs$_\mathcal{E}$↓ | FAD↓ | FD↓ | CDPAM$_T$↓ | CDPAM$_{mean \pm std}$ ↓ | CDPAM$_E$↓ |
|---|---|---|---|---|---|---|
| - | 0.073 | 10.38 | 56.02 | 1.052 | 0.085 ± 0.054 | 0.198 |
| 4 | 0.056 | 9.85 | 56.09 | 0.847 | 0.068 ± 0.045 | 0.178 |
| 8 | 0.059 | 10.01 | 56.35 | 1.035 | 0.073 ± 0.051 | 0.180 |
| 16 | 0.059 | 9.95 | 56.14 | 1.058 | 0.075 ± 0.052 | 0.169 |
| 32 | 0.130 | 10.77 | 59.06 | 0.818 | 0.058 ± 0.082 | 0.158 |

smaller inference steps. While larger inference steps appear to slightly improve reconstruction error in the music morphing task, however, in the case of the Laughing-Crying baby sound and Door knock-Clapping sound, the smaller DDIM steps result in a lower CDPAM$_E$ score. Thus, we cannot conclusively establish a strong relationship between inference steps and perceptual resynthesis perceptual error. One possible reason for results in music morphing task is the input audio music are synthesised by AudioLDM2 with 200 inference steps, therefore, larger inference steps helps for improving reconstruction quality in that case. Overall, larger inference steps indicates a slight improvement on morph quality cross the four sound groups in this experiment. However, larger inference steps require longer time consumption on binary serach with SPDP algorithm. Therefore, we suggest a trade-off between overall algorithm time-comsumming and morphing quality when setting the DDIM inference steps for SoundMorpher.

### 11.2 ABLATION STUDY ON MODEL ADAPTATION

In this experiment, we conduct an ablation study on model adaptation with LoRA on task of music morphing. We test SoundMorpher with different LoRA rank as well as SoundMorpher without model adaptation. Following Yang et al. (2023), we train LoRA parameters for 150 steps with 1e-3 learning rate. We also set unconditional bias correction with $r_0 = 2$ for 15 steps with 1e-3 learning rate. Table 8 shows the results of SoundMorpher with different rank size on model adaptation settings. According to Table 8, SoundMorher without model adaptation has obvious performance drop on morphing correspondence compares results with LoRA model adaptation. Even though higher LoRA rank has a slight improvement on perceptual reconstruction quality, however, SoundMorpher with $r = 32$ indicates poor correspondence with large MFCCs$_\mathcal{E}$ and large smoothness variance CDPAM$_{std}$. This result indicates that SoundMorpher with higher LoRA rank not lead to a better morphing quality. When $r = 4$, SoundMorpher achieves best performance on smoothness, and correspondence compared to $r = 8$, $r = 16$ and $r = 32$. Therefore, we suggest LoRA rank for model adaptation in SoundMorpher shouldn't be a large value such as $r = 32$.

In image morphing task by Yang et al. (2023), they observed that higher LoRA rank on model adaptation leading to more diverse image morping path. However, our results indicate different observation. One possible interpretation is, different from image morphing, diverse audio morphing path may lead to a large semantic gap, which resulting a higher FAD, FD and MFCCs$_\mathcal{E}$ (i.e., poor

Table 9: Ablation study on classifier-free guidance (CFG) scales on music morphing task

| CFG scale | MFCCs$_\mathcal{E}$↓ | FAD↓ | FD↓ | CDPAM$_T$↓ | CDPAM$_{mean \pm std}$↓ | CDPAM$_E$↓ |
|---|---|---|---|---|---|---|
| min: 1.5 - max: 3 | 0.150 | 10.95 | 59.20 | 0.808 | 0.058 ± 0.088 | 0.157 |
| min: 1.5 - max: 4 | 0.056 | 9.85 | 56.11 | 0.847 | 0.068 ± 0.045 | 0.178 |
| min: 1.5 - max: 5 | 0.055 | 9.81 | 56.01 | 1.101 | 0.078 ± 0.056 | 0.189 |
| min: 2 - max: 4 | 0.064 | 9.91 | 56.68 | 1.043 | 0.074 ± 0.053 | 0.173 |
| min: 3 - max: 5 | 0.068 | 9.96 | 57.46 | 1.074 | 0.076 ± 0.055 | 0.174 |

correspondence). This phenomena leads to a future study on how LoRA rank affects SoundMorpher performance in different morphing scenarios.

### 11.3 ABLATION STUDY ON CLASSIFIER-FREE GUIDANCE (CFG) SCALES

In this experiment, we explore impacts of CFG scales on SoundMorpher, we conduct an ablation study on music morphing task with $N = 15$ on different sets of max-min CFG scales in Table 9. According to our experimental results, maximum scale controls correspondence quality and smoothness quality of morphed results, whereas higher maximum scale leads to a lower MFCCs$_\mathcal{E}$ and higher CDPAM$_{mean}$± CDPA$_{std}$. In contrast, minimum scale controls intermediate quality of morphed results, where higher minimum scale leads to higher CDPAM$_T$.

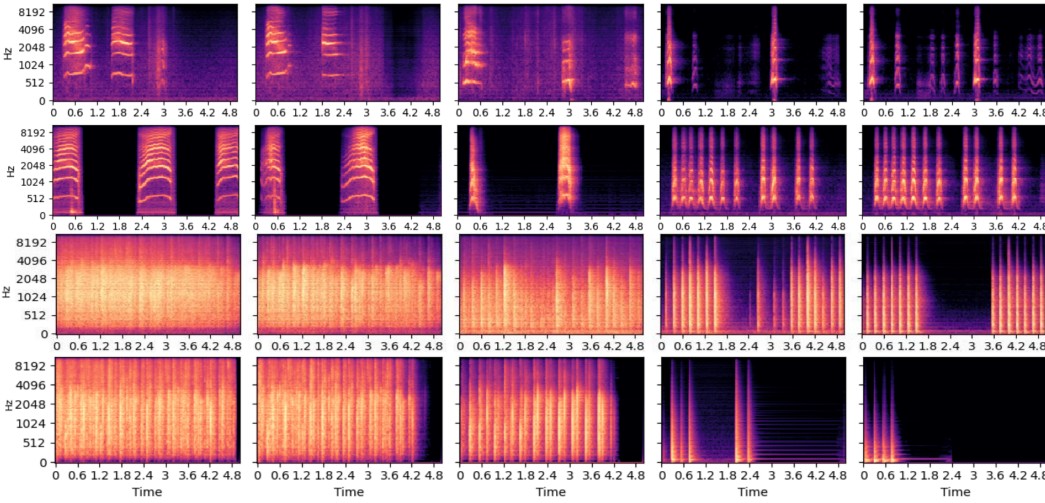

Figure 6: Failure cases for SoundMorpher with $N = 5$. The source and target sounds that have significant semantic difference in contents, this leads SoundMorpher to produce abrupt transitions.

### 11.4 FAILURE CASE

Although SoundMorpher produces high-quality sound morphing results, abrupt transitions can occur when the source and target sounds have significant temporal structure differences. A clear example of this is attempting to morph continuous environmental sounds with sounds that contain more silence. One obviouse example is to morph continuous environmental sounds and sounds contains more slience as Figure 6 shows.

Environmental sounds often consist of discrete and temporally separated events, such as a dog barking or a cat meowing, which have distinct and abrupt characteristics. These are inherently different from the more continuous and harmonically structured nature of music, where elements blend more fluidly over time. As a result, creating smooth transitions between such disjointed environmental sounds can be more challenging, leading to the perception of more abrupt or less natural transitions in the morphing process.

## 12 MORE VISUALIZATION EXAMPLES

This section provides more visualization examples for our experiment. Figure 7 provides additional visualization of timbre morphing compared with SMT under a paired piano-guitar music composition sample. Compare to SMT, SoundMorpher produces a smoother morphing that continuously connects target and source timbre points in the timbre space with closely spaced transition.

Figure 8 displays three examples of timbre morph with different musical instruments. SoundMorpher produces high-quality and smooth morphing with $N = 11$ perceptual-uniform intermediate morphed results.

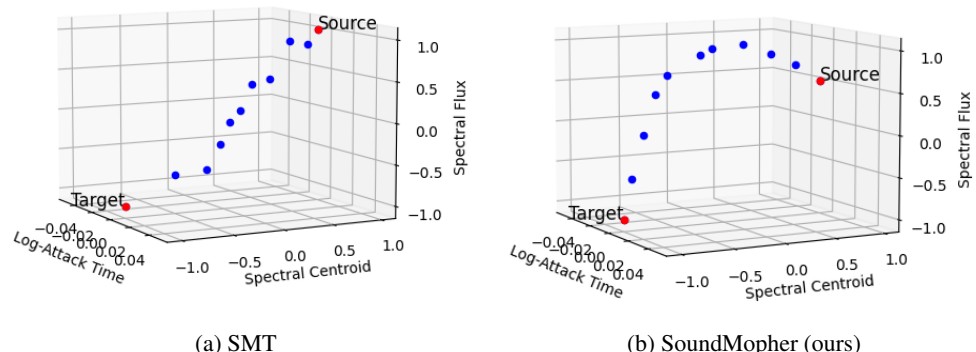

        (a) SMT                 (b) SoundMopher (ours)

Figure 7: Timbre space visualization of morph trajectories for piano-guitar timbre morphing. *Sound-Morpher produces a smoother and more continuous morph with closely spaced intermediate points.*

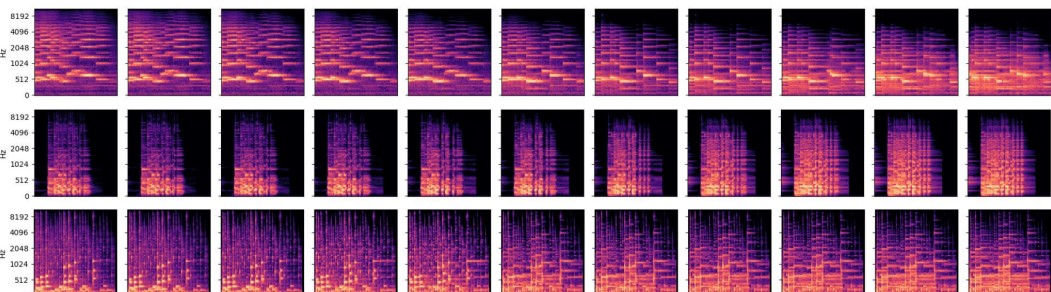

Figure 8: Visualization of timbre morphing for musical instruments with $N = 11$.

Figure 9 demonstrates visualization for environmental sound morphing experiment. This shows how SoundMorpher transitions between various environmental sounds, offering insights into the smoothness, quality, and intermediate stages of the morphing process.

Additionally, we randomly select audio samples from AudioCaps (Kim et al., 2019a) and use Sound-Morpher with $N = 10$ to perform complex sound scene morphing. Compared to the ESC50 dataset, the audio samples in AudioCaps often contain sound scenes involving multiple complex physical events. Visualizations of as shown in Figure 10.

These visualizations demonstrate that SoundMorpher effectively produces high-quality morphing across diverse audio types, including complex environmental sounds, music, and various musical instrument timbres. This highlights the flexibility and efficiency of SoundMorpher, showcasing its potential applicability in multiple real-world scenarios.

## 13 DATA SOURCE

This section contains details of the open-sourced data we used in our experiments.

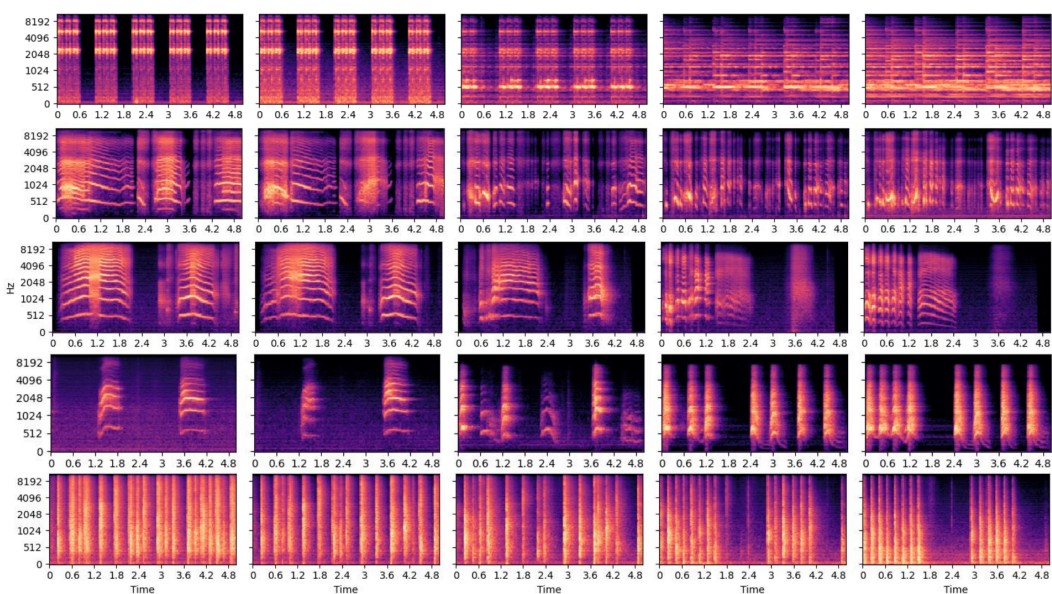

Figure 9: Visualization of environmental sound morphing with $N = 5$, from top to bottom: (1) church bells $\leftrightarrow$ clock alarm (2) crying baby $\leftrightarrow$ laughing (3) crying baby $\leftrightarrow$ laughing (4) cat $\leftrightarrow$ dog (5) clapping $\leftrightarrow$ wood door knocking

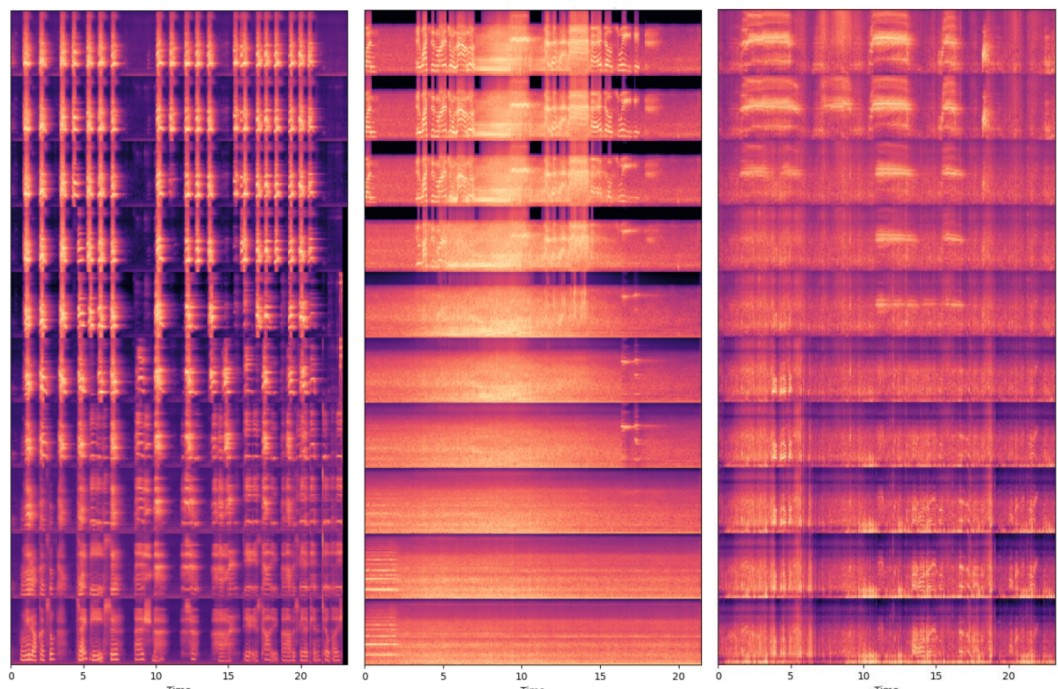

Figure 10: Visualization of complex sound scenes from AudioCaps by SoundMorper with $N = 10$.

### 13.1 TIMBRAL MORPHING FOR MUSICAL INSTRUMENTS

- 8 pairs of piano-guitar and 8 pairs of harp-kalimaba audios: https://harskish.github.io/Timbrer/index.html.

- 6 pairs of timbral transfer audios with isolated musical instruments:

https://wry-neighbor-173.notion.site/MusicMagus-Zero-Shot-Text-to-Music-Editing-via-Diffusion-Models\
-8f55a82f34944eb9a4028ca56c546d9d.

## 13.2 SOUND MORPHING

- 100 pairs of randomly selected environmental sound effects from ESC50 dataset: https://github.com/karolpiczak/ESC-50

## 13.3 MUSIC MORPHING

- 21 musical compositions with different instruments and genres: https://audioldm.github.io/audioldm2/.

## 13.4 MODEL COMPARISON WITH MORPHFADER

- 7 pairs of sound examples compared with MorphFader (Kamath et al., 2024): https://pkamath2.github.io/audio-morphing-with-text/webpage/audio-morphing.html

