# OpenReview forum: "SoundMorpher: Perceptually-Uniform Sound Morphing with Diffusion Model"
_ICLR.cc/2025/Conference — ICLR 2025 Conference Withdrawn Submission_

### Official Review · Reviewer_3zUQ · 2024-10-31

**Soundness:** 3
**Presentation:** 2
**Contribution:** 3
**Rating:** 5
**Confidence:** 4

**Summary:**

The paper presents a MusicLDM2-based inference-time method (requiring no model training or fine-tuning) for morphing various types of music and sounds across scenarios such as timbral morphing of musical instruments, environmental sound morphing, and music morphing. The authors claim their approach differs from traditional sound morphing methods, which assume a straightforward, linear relationship between the morph factor and perceived sound qualities. Traditional methods oversimplify the complexity of sound perception and limits morph quality. In contrast, they propose SoundMorpher, which establishes a more accurate, proportional mapping between the morph factor and the perceived characteristics of morphed sounds using Mel-spectrograms. SoundMorpher employs the Sound Perceptual Distance Proportion (SPDP) ensuring that changes in the morph factor correspond uniformly to perceptual changes. This approach enables smoother sound transitions and maintains consistent quality across different sound-morphing tasks.

The paper also introduces metrics to evaluate sound morphing systems based on three key criteria: correspondence, perceptual intermediateness, and smoothness. Extensive tests demonstrate SoundMorpher's effectiveness and adaptability, with potential applications in music creation, film post-production, and interactive audio technologies.

Demo audio files are provided in the supplementary material.

**Strengths:**

This is a fairly well-written paper. I found it interesting and a pleasure to read. Even though the method it presents is not a groundbreaking novelty and is only an inference-time "trick" to achieve good sound morphing, I found it a clever way of handling and extracting desired results from a pre-trained generative model. The related work section is quite comprehensive (at least, I couldn’t recall any paper that hasn't been mentioned here), the objectives of the work are clear, and the method is fairly well explained. The proposed metrics do make sense (I’ll come back to this in the weaknesses section), and the algorithms (show in the appendix) and descriptions of the methodology are quite comprehensive and help understanding.

**Weaknesses:**

Here is the list of my concerns following the order of the sections.

Sound morphing preliminary:

In the paragraph of line – 141 -149, authors define 3 criterias Correspondence, Intermediateness, and Smoothness. I think there are lots of critiques one can brung up to these criterias.

1.	Correspondence: This criterion requires that the morph captures semantic-level transitions. However, perception of “semantic” qualities in sound can be subjective and context-dependent. If listeners interpret these qualities differently, then it may be difficult to ensure correspondence consistently across diverse types of sounds, especially if the morphing involves highly variable sources, like environmental sounds versus musical instruments.

2.	Intermediateness: This criterion expects listeners to perceive intermediate morphs as “between” the source and target sounds. However, the perceptual “in-betweenness” may not always be straightforward in audio and can be (again) subjective. If sounds have distinct timbral qualities, they might not blend smoothly. In such cases, perception could be binary or categorical(?), with certain morphing points perceived as “closer” to one sound or the other rather than as true intermediates.
For example, in the audio demo files, there is a cat-to-dog morph. Sometimes, the cat's meow gradually changes into a dog’s bark, but at other times, there is an intermediate sound containing both the cat and dog sounds (like an audio scene where both animals are present, which is quite realistic and could be perceived as intermediate). So, which of these is truly intermediate? Or are both? I don’t see a clear description addressing this. Please correct me if I’m missing something.

3.	Smoothness: This criterion assumes that a linear change in the morph factor will correspond to a consistent, linear perceptual transition. I have a major question here: why is a linear change necessarily good? As we know, sound perception does not always follow linear increments; auditory characteristics like volume and pitch are often perceived on logarithmic or exponential scales. This means that, depending on where we are in the spectrum, even small changes in the morph factor might lead to abrupt perceptual shifts, while large changes may go unnoticed by human listeners. This would violate the criterion of smoothness, correct?
For example, if a bass instrument is morphing into a high-pitched violin and pitch is increasing by a fixed step (say 100 Hz), that change at a low frequency would be perceived as quite large, while at higher frequencies (around 2000 Hz), the same 100 Hz step wouldn’t be perceived as a significant change. So, what exactly do the authors mean by "linear change"? when they say linear in context of perceptual doe this mean logarithmic in metrics? I didn’t find a clear explanation on this in the paper but this is a major question, I think, and need to be mentioned and clearly explained.

Evaluation:


My main concern with this paper lies in the experiments and evaluation section. The issue I see is that the work develops its own metrics and then uses these motly for just reporting the number of the model. For the comparison with baseline(s) same metrics are used, rather than relying on established metrics (at least thiose used in the prevous work for fair comarison). This approach makes comparisons with existing work somewhat shaky.

Here are the details.

1.	Evaluation metrics

Lines 1021 and 1023-1024: Did you mean “consecutive” instead of “consentive”?
The calculation of FAD and FD is unclear. Are these values averaged? The paragraph starting at line 1019 is not entirely clear. I see there are instrument groups, but it’s not fully explained how FAD and FD are calculated along the morphed path. Sounds in the middle of this path would presumably be intermediate between the two instruments. Are we calculating FAD and FD across the entire sequence and then averaging, or only at the midpoint? If it's the midpoint, what are we comparing it to—the source or the target? I’m unsure what FAD and FD are meant to reveal in this context. Could you provide a more detailed explanation, maybe including a formula for how final FAD is calculated in the tables (lines 270, 380, and 447)?

Lines 316-320: use of CDMAP seems not fully justified.  CDMAP is typically used for speech. Is it suitable for measuring instrument and environmental sounds? Could you provide some justification for why this metric is relevant here?

2.	TIMBRAL MORPHING

The baseline model chose here is weird the Sound Morphing Toolbox (SMT) from Caetano, 2019. This is very old and not a ML model. Why don’t authors use some more recent work for the comparison. In the demo files the files generated by SMT sound terrible! I don’t think SMT is even worth considering as a baseline. I’m sure much better quality of audios can be achived by usieng any other concurent ML based baseline.

3.	ENVIRONMENTAL SOUND MORPHING

No baseline, Just showing the numbers for their own metrics.

4.	MUSIC MORPHING

Same concern here, no baseline! I don't undestand what are are these numbers say without a baseline...

5.	Subjective Evaluation:

Again, for some reason, the subjective (human) evaluation does not compare the proposed method with other methods.

Generally, in the paper, baseline models are not used in the evaluation. The paper only presents results for the proposed model, with the only comparison being in timbral morphing for musical instruments—and even that is with an older signal processing method, the Sound Morphing Toolbox (SMT) from Caetano, 2019. The related work section references numerous concurrent works in the field, so I don’t understand why the authors did not adopt any of these as baselines.

6.	Model comparison

In the "Model Comparison" subsection at line 472, the authors compare their method to a concurrent work, MorphFader (Kamath et al., 2024), but they only use the 7 examples provided on the demonstration page of that work. I don’t believe that 7 examples are sufficient for a robust comparison. I would have liked to see a comparison with this model (or other concurrent models) across all scenarios with all metrics and bigger test dataset, including those used by the baseline models, for a fair assessment. Additionally, they report only CDPAMT, CDPAMmean±std, and MFCCsE—why limit the metrics?




There are some small typos and grammatical errors that I spoted:

In the line 013: if I understand what authors say correctly, "models" should be "model" to match the plural subject "methods."

Line 243 there is a typo: adaption->adaptation.

It would be nice to have table 1 on the page where it is dsuicussed.

Line 444-445, in the table 3 N is number of components of PCA? This is misleading because N was used for N a number of sounds in morphed sequence between sores and target lines (190—193).

The experiment section is structured a little weird. I don’t understand why in experiments (where we also have results presented by scenarios) we have subsection “discussion”. And this subsection presents Mean opinion score, which, in my opinion, is the most valuable part of results section.

Line 1207 – instead of “achieves” must be “be achieved”, I suppose.

**Questions:**

Questions and Suggestions for Improvement

From the comments above here are some questions/suggestions for the authors:

Sound Morphing Preliminary:

o	Correspondence: few words about subjectivness of the semantic meaning would help here I think.

o	Intermediateness: The example I gave in the comments above, the cat-to-dog morph in the demo files sometimes produces an "in-between" sound that contains both the cat and dog sounds, which might be realistic but isn’t necessarily an intermediate. Could the authors clarify what qualifies as intermediate? Are both interpretations valid?

o	Smoothness: Could the authors explain what they mean by "linear change"? Although numbers are used (Mel spectrograms in this case), it’s still not fully clear. For elements like pitch and loudness, does this “linear change” involve logarithmic scaling in metrics? Is it based on human perceptual linearity, and if so, what exactly does that mean, and how does it translate to audio metrics? A more explicit explanation would be helpful. In lines 258–265, the authors state that the Mel spectrogram is a good approximation of human perception, but I think a few more words on the logarithmic nature of audio perception and how the Mel scale assists here would strengthen the explanation.

________________________________________
Evaluation Concerns:

2.	Evaluation Metrics:

o	Please provide exact formula of how FAD and FD are calculated or please direct me to where it is presented.

o	Could you provide justification for why CDPAM is relevant metric?

3.	Timbral Morphing:

o	The choice of baseline model (Sound Morphing Toolbox from Caetano, 2019) seems outdated and not representative of current ML methods. Could the authors consider more recent ML-based baselines? This would provide a fairer comparison, especially as the SMT-generated audio files seem quite low quality.

4.	Environmental Sound Morphing:

o	No baseline model is used here, and the results are presented only with the authors' own metrics. Could the authors incorporate a baseline to add context to their results?

5.	Music Morphing:

o	Similarly, no baseline model is used here. Without a baseline, it is difficult to interpret the significance of the reported results. Could the authors add a comparison baseline to strengthen the evaluation?

6.	Subjective Evaluation:

o	In the subjective (human) evaluation, no comparison is made with other methods. Why did the authors choose to evaluate only the proposed model, especially considering the number of concurrent works mentioned in the related work section? Adding more baseline comparisons would enhance the evaluation's rigor.

7.	Model Comparison:

o	In the "Model Comparison" section (line 472), would it be possible to expand this comparison across all scenarios, including metrics from MorphFader or other models, for a fairer and more comprehensive comparison? Additionally, the authors report only CDPAMT, CDPAMmean±std, and MFCCsE. Why were these metrics chosen, and why were other metrics not included?

---

> ### Author Response · Authors · 2024-11-15
>
> We thank the reviewer for appreciating our work and the time you spent on the paper. Your insightful suggestions have been invaluable in enhancing the quality and clarity of our paper. We hope that our response will address your concerns. We are happy to respond to additional questions during the discussion period.
>
> **1. About sound morphing preliminary in Section 3.1**
>
> We thank for the insightful suggestion about the clarity of the three criterions, we will revise this section with more details.
>
> The three objective criterions are originally come from Caetano \& Osaka (2012), which are not proposed by us (see l.67-68, 141-142, 533-534). Our contribution and motivation is to adapt a series of comprehensive quantitative metrics according to the criterions to further provide a formal evaluation system for future sound morphing comparison. We hope below can address your concern about the clarity of the three objective criterions.
>
> _Correspondence_: According to Caetano \& Osaka (2012): 'morphing requires a description of the entities being morphed (shapes, images, sounds, etc), followed by establishing the correspondence between these descriptions'. This criterion focuses on the semantic content (or description) of audio rather than perception. For example, in music morphing, the correspondence may ensure the morphed outputs have description in between source and target music, rather than an entire new music. In environmental sound morphing, semantic content may represent the class of environmental sounds.
>
> _Intermediateness_: As in Caetano \& Osaka (2012), the intermediateness criterion illustrates the morphed objects should be perceived as intermediate. The most straightforward way to evaluate the intermediateness through a morphed sequence, where if the perceptual difference between the current transition and the previous transition is large, the large perceptual jumps may cause the intermediate objects to deviate from their role as "in-between" states, violating the intermediateness criterion. The "in-between" state could be interpreted by both the cases, which are interpreted as conceptual and categorical perception by Caetano \& Osaka (2012). Sound is different from images or shapes. Sound morphing can draw inspiration from visual techniques, such as smooth transitions and perceptual metrics, but must adapt these to the auditory domain.
>
> _Smoothness_: The linear assumption assumes each transition should have the same amount of perceptual change (i.e., $\Delta p$) if change the same amount of morph factor (i.e., $\Delta \alpha$) (see l.146-147). This means the relationship between morph factor and morphed sound perception is linear (i.e., you can image a linear function where x-axis is morph factor and y-axis is morphed sound perception). It can be formulated as
> $ x^{(\alpha_{i+1})} - x^{(\alpha_{i})} = p_{\alpha_{i+1}} - p_{\alpha_{i}} = C$, where $C$ is a constant.
>
> **2. Regarding the Mel scale and human perception.**
>
> We thank for this insightful suggestion. We actually used the log-scaled Mel spectrogram in Equation(7), which potentially solved the problem of imbalance change in low-frequency and high-frequency areas. We will clarify the connection of the Mel scale with the nature of human perception in our revised version, with proper references.
>
> **3. The comparisons of this paper are based on its own metrics.**
>
> Since existing works target on a specific sound morphing scenario, in which metrics they used only applicable under the certain scenario. According to Caetano & Osaka (2012), Gupta et al., (2023) and Kamath et al., (2024), existing sound morphing methods lack of a formal and comprehensive evaluation system. For example, MorphFader (Kamath et al., 2024)performs morphing by text prompt, which evaluates ‘smoothness’ by text-audio similarity scores based on CLAP. MorphGAN (Gupta et al., 2023) performs audio texture morphing by interpolating parameters in an audio classifier, in which calculates parameter sensitivity and distribution similarity.
>
> Instead, our motivation is to address this problem and make a formal sound morphing evaluation system that may benefit to future comparison. Furthermore, the comparison with MorphFader already showcases its advantage of flexible and formal evaluation, even if MorphFader doesn't release their codes, we can still make a fair comparison with the provided morphed results.

---

> ### Author Response · Authors · 2024-11-15
> **Response cont.**
>
> **4. Detail formulation for computing FAD and FD in the experiment.**
>
> The calculation of FD and FAD are similar. Given two sets of audio data, $X_{morph}$ and $X_{sourced}$, where $X_{morph}$ contains consecutive morphed results as $X_{morph} =\\{x^{\alpha_i}\\}^N_{i=1}$, and $X_{sourced}$ contains sourced audios in that class.
>
> Step 1. Calculate feature embeddings by a pre-trained audio classification model (i.e. VGGish for FAD and PANNs for FD), where embeddings capture high-level features of the audio.
>
> Step 2. Calculate the mean $\mu$ and covariance matrix $\Sigma$ of the feature embeddings for both $X_{morph}$ and $X_{sourced}$. As $X_{morph} \sim N(\mu_{morph},\Sigma_{morph})$ and $X_{sourced} \sim N(\mu_{sourced},\Sigma_{sourced})$.
>
> Step 3. metric = $|| \mu_{sourced} - \mu_{morph}||^2 + Tr(\Sigma_{sourced} - \Sigma_{morph} - 2\sqrt{\Sigma_{sourced}\Sigma_{morph}})$
>
> Since the mean and covariance matrix captures distribution information of the morphed and sourced audio sets. Therefore, this metric helps to evaluate semantic consistence between morphed results and sourced samples. The smaller value of the metrics represents they are in a closer distribution level. In our experiment, we use the package from https://github.com/haoheliu/audioldm_eval to obtain the two metrics.
>
> **5. Could you provide justification for why CDPAM is a relevant metric?**
>
> CDPAM is trained to align with human perception, focusing on exploring higher level of perceptual audio features. Music and environmental sounds, like speech, rely on perceptual qualities (e.g., timbre, rhythm, spatial texture) for evaluation, which makes CDPAM conceptually extendable. Although CDPAM is evaluated on speech related tasks in their paper, its focus on contrastive learning of perceptual features often translates well to other domains where such features are important. For instance, music involves harmonics and temporal dynamics, similar to speech prosody. Environmental sounds rely on texture, temporal patterns, and frequency richness, overlapping with features modeled by CDPAM.
>
> There are some references that use CDPAM for evaluating environmental sounds and music data:
>
> [1] Hai, Jiarui, et al. "Dpm-tse: A diffusion probabilistic model for target sound extraction." ICASSP 2024-2024 IEEE International Conference on Acoustics, Speech and Signal Processing (ICASSP). IEEE, 2024.
>
> [2] Jacobellis, Dan, Daniel Cummings, and Neeraja J. Yadwadkar. "Machine Perceptual Quality: Evaluating the Impact of Severe Lossy Compression on Audio and Image Models." arXiv preprint arXiv:2401.07957 (2024).
>
> **6. Lack of comparison with baselines for timbral morphing, environmental sound morphing and music morphing.**
>
> Please see Q3 in official comments and Q3-5 for the rebuttal response of Reviewer HAc4.
>
> Even though SMT is an out-of-date timbral morphing method; however, we can't find any other open-sourced related methods to make comparison in the task of timbral morphing. As we mentioned in l.40-41, the sound morphing methods based on traditional signal processing techniques limits application for inharmonic sounds. Therefore, it is not surprising that the results from SMT are not performing well on complex music compositions. We also found SMT fails in the case of Taiko-Hihat in our experiment (see l.366-367).
>
> Unfortunately, we cannot find any ML-based similar methods mentioned in our related work section with official implementation codes for a direct and fair comparison for the three tasks. We tried our best to find a fair and direct comparison with other existing methods, that is the reason why we also compared with a concurrent method based on examples from their demo page even though they haven't released their codes yet.
>
> If the reviewer has any suggestions about the baseline methods, please elaborate.
>
> **7. Model comparison with the concurrent work**
>
> We thank for the suggestion from the reviewer, unfortunately, MorphFader hasn't released their implementation codes. We cannot do more rather than making direct comparison with the 7 samples on their demo page.  We provide further explanation about this model comparison as well as a visualization comparison in Appendix 10. As we claimed in Appendix 10, we cannot compute FAD, FD, and CDPAM$_{E}$ value since the demo page doesn't provide actual sourced audios rather than reconstruction audios when $\alpha= 0$ and $\alpha =1$.
>
> **8. Regarding MOS study**
>
> Due to the lack of direct fair comparable baseline method in sound morphing research, this MOS study aims to validate the robustness of SoundMorpher across two different real-world applications, and further emphasis the robustness of a generalized sound morphing method.
>
> **9. Typos and the repeated notation (i.e., $N$ in PCA) in the paper.**
>
> We sincerely thank for the careful review, we will correct the these accordingly.

---

### Official Review · Reviewer_kD4m · 2024-11-03

**Soundness:** 3
**Presentation:** 3
**Contribution:** 1
**Rating:** 3
**Confidence:** 5

**Summary:**

The authors present a method to morph sounds in an intended way.

They rely on a pretrained AudioLDM2 model (a latent diffusion model for sounds and music) and propose to generate morphed sounds by interpolating between two sounds.

"Textual inversion" is performed in the condition embedding level for each sound to obtain embeddings $E$.
Latent representations $z_T$ are obtained using the probability flow ODE.
Morphed sounds are obtained by generating using the probability flow ODE from $(z_T^\alpha, E^\alpha)$ where the $z_T^\alpha$ is a slerp interpolation of the $z_T$'s and $E^\alpha$ a linear interpolation of the $E$'s.

Binary search is then performed on this trajectory parameterized by $\alpha$ to obtain samples equally-spaced according to some perceptual measure, relative L2 norm over mel spectrograms in the present case.

**Strengths:**

The paper is well-written and easy to follow, with examples provided in the supplementary material.
There is an extensive experimental part with a user study, but mainly compares with SMT.

**Weaknesses:**

- The paper lacks novelty. Most of the elements presented in the paper do come from
Yang et al.2023 IMPUS: IMAGE MORPHING WITH PERCEPTUALLYUNIFORM SAMPLING USING DIFFUSION MODELS.
The present paper can thus be seen as a straightforward adaptation of IMPUS to the audio domain.
- If LPIPS was chosen in IMPUS as the perceptual metric, here the choice of L2 over mel-spectrograms may be less appropriate.
- Concatenating audio segments in Eq. 8 in x-space seems to produce abrupt transitions. Please not that with most of the architectures used for diffusion models, it would be possible to have this chunk-based generation done directly in z-space.

- There are some missing references concerning controlled interpolations using audio diffusion models.
- Provided audio examples are not particularly convincing and transitions sound abrupt, especially with as few points as five.

**Questions:**

- Where do the "Static morphing; Cyclostationary morphing & Dynamic morphing" terms come from?
- Why not using CDPAM as the perceptual metric?
- The choice of the interpolating path is pretty arbitrary, and the presented technique could be applied over any path. Is the perceptual metric always monotonic over these trajectories?

---

> ### Author Response · Authors · 2024-11-15
>
> We extend our sincere thanks to the reviewer for dedicating their time to review our paper. We hope our response can address your concerns and make it clear to you on our motivations and contributions. We are happy to respond to additional questions during the discussion period.
>
> **1. The paper lacks novelty, and it is a straightforward adaptation of IMPUS to the audio domain.**
>
> Please see Q1 and Q2 in the official comment.
>
> **2. The choice of L2 over mel-spectrograms may be less appropriate**
>
> As we explained in l.257-262, Equation(7) elegantly solves the challenges we mentioned in l.252-254. This solution is also been agreed by Reviewer 3zUQ as well has been proved by our ablation study in Section 5.5. If the reviewer insists the Mel-spectrogram in Equation(7) is less appropriate, please elaborate.
>
> **3. Why not using CDPAM as the perceptual metric?**
>
> As we claimed in Section 3.1 as well as in l.258-259, the desired sound morphing results should not only ensure perceptual intermediateness and smoothness, but also should ensure correspondence. **These three criterions are proposed by Caetano \& Osaka (2012) to formalize the evaluation of sound morphing.** Although CDPAM provides perceptual distance between two audios, we cannot ensure the $x^{(\alpha_i)}$ in-between $x^{(0)}$ and $x^{(1)}$ if using CDPAM as the metric in Equation(7). **This violates the intermediateness criterion.** Additionally, CDPAM doesn't provide semantic (e.g., audio description) information of audio, which **cannot ensure the correspondence criterion**. Rather, Mel-spectrogram provides both perceptual information and description of an audio, which is a better solution for the three criterions.
>
> **This strongly highlights that our method is not merely a straightforward adaptation of IMPUS but incorporates meaningful insights specific to the sound morphing domain.**
>
> **4. Concatenate z-space in Equation(8).**
>
> This is an insightful idea. Concatenating in the z-space for dynamic morphing indeed produces better audio quality compared to concatenating waveforms. While waveform concatenation does not exhibit obvious abrupt transitions and offers advantages such as controllability in the time dimension, we will revise Equation (8) to provide the alternative solution of z-space concatenation. We thank for this valuable suggestion.
>
> **5. There are some missing references concerning controlled interpolations using audio diffusion models.**
>
> We tried our best to provide a comprehensive literature review for the sound morphing task. This is also been agreed by Reviewer 3zUQ. If the reviewer has any suggestions, please elaborate.
>
> **6.Provided audio examples are not particularly convincing and transitions sound abrupt, especially with as few points as five.**
>
> As we mentioned in our demo page, "Our demonstration also includes failure cases with abrupt transitions for SoundMorpher that two input sounds as significant semantic differences in content" to further illustrate the limitation that we observed in Appendix 11.4 (please also see l.522 - 525 in main text).
>
> **7. Where do the "Static morphing; Cyclostationary morphing \& Dynamic morphing" terms come from?**
>
> The three morph methods are commonly used in the sound morphing, as we introduced in Section 3.1. We humbly suggest the reading group video recording of sound morphing in https://www.youtube.com/watch?v=fI8Wqc7e3Zk (start from 26 min 30 sec) may help the reviewer to understand the three sound morphing methods.
>
> **8. The choice of the interpolating path is pretty arbitrary, and the presented technique could be applied over any path.**
>
> We don't understand the meaning of "arbitrary". If we have misunderstood, we kindly ask for clarification for this review to ensure we address your concern accurately.
>
> We interpolate conditional embeddings via a linear interpolation (l.232) as the conditional embeddings in AudioLDM2 are semantic structured, and we use spherical linear interpolation (l.238) to interpolate latent states following by Song et al. (2020). We perform perpetual uniform sound morphing by interpolating N points between $p^{(0)}$ and $p^{(1)}$ to ensure $\Delta p$ is constant in the morph path (l.269 - 282).
>
> **9. Is the perceptual metric always monotonic over these trajectories?**
>
> The proposed perceptual metric SPDP **strictly and always** monotonic over the morph trajectories, since it obtained by calculating the distance proportion between the Log magnitude Mel-spectrogram between source and target audio as in Equation(7). Given a SPDP point $p_i = [\tilde{p}_i^0,\tilde{p}_i^1]$, the summation of $\tilde{p}_i^{0}$ and $\tilde{p}_i^{1}$ is always equal to 1. Please also see Figure 8 - Figure 10 in Appendix 12 for a strongly visual evidence, where the spectrogram of morphed results are strictly in between source and target.
>
> **10. The experiment mainly compares with SMT.**
>
> Please see Q3 in the official comment.

---

### Official Review · Reviewer_HAc4 · 2024-11-03

**Soundness:** 2
**Presentation:** 3
**Contribution:** 2
**Rating:** 5
**Confidence:** 3

**Summary:**

This paper proposes a new method for sound morphing, defined as the generation of a sequence of perceptually intermediate sound samples for a given pair of source and target audio. As this problem is not very common in the audio processing community, the authors introduce different formulations of the problem (e.g., static vs. dynamic morphing) and discuss the criteria for a good morphing path, including correspondence, intermediateness, and smoothness. The authors propose an original method of sound morphing based on interpolating latents within AudioLDM2, demonstrating impressive results compared to the baseline. They also introduce several heuristics to make the transitions between consecutive samples along the morphing path more uniform and stable, including LoRA fine-tuning of AudioLDM2 and finding an optimal sequence of interpolation coefficients using binary search. Finally, the authors describe various experiments that demonstrate the strong performance of their model across different scenarios and audio domains.

**Strengths:**

- The proposed method is very reasonable and has already shown good results in image processing;
- Mostly, the quantitative comparisons are solid and show the effectiveness of the method in various audio domains;
- The paper proposes metrics to measure the morphing quality in terms of correspondence, intermediateness and smoothness which is important for further development of this research topic.

**Weaknesses:**

- The contribution of the paper is relatively incremental as the paper borrows most of its ideas from (Yang et al. 2023). All essential features of the proposed method such as latents interpolation inside a Latent Diffusion Model, LoRA adaptation, finding the optimal trajectory with binary search on a sequence of values of an auxiliary metric, and even the introduction of 3 metrics for model evaluation, were proposed in (Yang et al. 2023). The paper is basically an attempt to adapt the approach introduced in (Yang et al. 2023) to audio domain. The main challenge the authors face in this study is related to tuning AudioLDM 2 and designing perceptual metrics as in (Yang et al. 2023).
- Comparison with only one baseline on a music domain given in Table 1 seems insufficient to provide full understanding of the performance of the method. Although objective metrics and the recordings given in the supplementary material demonstrate huge improvement over the baseline, it more likely suggests that the baseline may be too weak or poorly tuned.
- The MOS scores given in Table 4 don’t have any grounding such as GT intermediate samples or a baseline which makes them uninterpretable. Moreover, because of a small number of assessors, the variance of the score is too large.
- Table 5 shows no improvement over the baseline in 2 metrics of 3. Intuition of the qualitative difference corresponding to the improvement of 0.04 in the first metric is not provided. Some illustrations of spectra and/or audio samples in the supplementary material corresponding to different levels of CDPAM may help here.
- Metric MFCC_Se introduced in Appendix 8 doesn’t evaluate the portion of the content from x_{source} and x_{target} but similarity of the spectra. The metric is easily minimized when we average x_s and x_t.
- Minor issues: “Spectral contras” p.9:450; “Mean opinion socre” p.20:1034

REFERENCES
Zhaoyuan Yang et al. IMPUS: IMAGE MORPHING WITH PERCEPTUALLY UNIFORM SAMPLING USING DIFFUSION MODELS, ICLR 2024

**Questions:**

The recordings given in the supplementary material are either midi or short isolated environmental sounds which raises concerns about applicability of the method to real data. Did you try to apply your method to real music recordings?

---

> ### Author Response · Authors · 2024-11-14
>
> We extend our sincere thanks to the reviewer for dedicating their time to review our paper. We hope our response can address your concerns and make it clear to you on our motivations and contributions. We are happy to respond to additional questions during the discussion period.
>
> **1. Regarding contribution and novelty**
>
> Please refer to Q1 in the official comment and Q3 in the response of reviewer kD4m. Our challenges are not only tuning a model and designing perceptual metrics; however, as we claimed in Section 3.1, sound morphing is inherently different from image morphing. Sound is more abstract than image and has various application scenarios, as our experiment illustrates.
>
> **2. Regarding three criterions that similar to IMPUS**
>
> Please refer to Q2 in the official comment.
>
>
> **3. Regarding the baseline with poor performance for timbral morphing**
>
> Even though SMT is an out-of-date timbral morphing method; however, we can't find any other open-sourced related methods to make comparison in the task of timbral morphing.
>
> As we mentioned in l.40-41, the sound morphing methods based on traditional signal processing techniques limits application for inharmonic sounds. Therefore, it is not surprising that the results from SMT are not performing well on complex music compositions.
>
> If the reviewer has any suggestions for a fair baseline comparison, please elaborate.
>
>
> **4. Regarding the MOS study with a small number of assessors**
>
> We obtained 21 completed survey answer sheets for the MOS study, which is a fair number according to other related works such as Kamath et al. (2024) (i.e., 18 assessors) and Zhang et al. (i.e., 26 assessors). Since there is no direct fair baseline methods for comparison, this MOS study aims to validate the robustness of SoundMorpher across different types of sounds, and further emphasis the robustness of a generalized sound morphing method.
>
>
> **5. Regarding the comparison with the concurrent work**
>
> We thank the insightful suggestion from the reviewer, we have already provided detail explanation for this comparison in Appendix 10 as well as visualization of spectrogram comparison in Figure 5. Again, what we want to emphasize is that the comparison is based on a concurrent work with improvement. Please also refer to Q4 in the official comment.
>
>
> **6. Regarding MFCC$_{\mathcal{E}}$ metric**
>
> The MFCC$_{\mathcal{E}}$ metric measures content consistency (i.e., the correspondeness criterion proposed by Caetano \& Osaka (2012)) in the midpoint of morphed sequences and is strictly obtained by the equation we provided in Equation (15). We didn't take an average, but we obtain the value by calculating the absolute error between the proportion of how the midpoint result with $x_s$ with 0.5. Since **it's a proportion metric not a similarity**, the summation of the morphed result between $x_s$ and $x_t$ is always equals to 1.
>
> **7. Minor issues: “Spectral contras” p.9:450; “Mean opinion socre” p.20:1034**
>
> We will correct the typo accordingly, thank you for the carefully reviewing.
>
> **8. Did you try to apply your method to real music recordings?**
>
> We thank for the insightful suggestion. Real music recordings may encounter problems such as copyrights and recording qualities. Instead, we will release our codes upon the publication for interested users. Evaluating on high-quality synthesized music is a commonly used method for music-related research, e.g., Zhang et al. (2024) and Kemppinen P., (2020).
>
> We provided a visualization of applying our method to complex sound scenes from AudioCaps in Figure 10. As well as the comparison with the demo examples from the concurrent work, MorphFader, is based on recordings from AudioPairBank. Due to the size limitation for the supplement, we cannot provide those in our demo page.
>
> This already sufficiently showcased our method is applicable in the real-world applications.

---

> ### Comment · Reviewer_HAc4 · 2024-11-22
>
> p3. "If the reviewer has any suggestions for a fair baseline comparison, please elaborate."
>
> Sure. In this case, you might have reproduced any of the sota papers in your domain. It is, of course, not an easy task but it would make your results look stronger. In the current situation, the reviewer cannot properly assess your results as the authors: 1) don't have propoper baseline; 2) don't provide convincing demo samples.
>
> p4. Lack of baseline and anchoring (e.g. adding a deliberately bad and/or gt recording) given the difficulty of the concept of 'intermediateness' makes the MOS scores practically meaningless. It doesn't tell us that the model performs 'equally well' in both cases but that the score is volatile and random (see variance).
>
> p6. MFCC is a decorrelated and compressed (PCA-like) representation of the mel-spectrum. I really don't understand what you mean by content here.

---

> ### Author Response · Authors · 2024-11-25
>
> We thank for the reviewer's feedback and the effort you spent on our paper. We wish below response can further clarify our paper. We are happy to respond to additional questions during the discussion period.
>
> **1. To reproduce any of the SOTA papers in your domain for a comparison.**
>
> We appreciate the reviewer's suggestion; however, we respectfully disagree
>
> (1) Even though the lack of direct and open-sourced comparison in the sound morphing area, we will release our implementation codes upon the publication, which our paper will be the first to provide a fair and direct comparison for future sound morphing studies.
>
> (2) We have already compared with a concurrent work, which is superior to any public SOTA methods, this comparison already showcases our method is even superior to the concurrent method.
>
> (3) Reproducing other SOTA papers is a solution; however, it is not a fair comparison. The papers may not offer enough detail for reproducing, and the dataset they used is not always accessible. As the reviewer suggests, low-quality results are not convincing.
>
> **2. Don't provide convincing demo samples and add a deliberately bad recording.**
>
> We thank for the reviewer's feedback; however, we humbly and respectfully disagree with this. We emphasize that, as we claimed in our demo page, our demo samples are without cherry-picking. The abrupt samples also provided on the demo page and discussed as limitation in Appendix 11.4. We don't understand what does the 'adding a deliberately bad recording' means. **Both the baseline and the dataset we used are open public.** The baseline method is open-sourced in https://github.com/marcelo-caetano/sound-morphing, and we have provided data source details in Appendix 13.
>
> **3. MOS study does not tell use the model performs equally well.**
>
> We have provided how the MOS study designed in Appendix 9, we designed this subjective study to verify the morphed results according to the three criterions. Although the variance of MOS score is relative higher, sound morphing is different from speech and sound generation tasks, as the sound morphing has no ground truths. This may cause the subjective study with higher variance; however, the mean of MOS scores for the two tasks are at the similar level. In addition, our demonstration also provides evidence that our method is successfully applied to either music morphing or environmental sound morphing.
>
> **4. Regarding the MFCC metric.**
>
> The MFCC$_{\mathcal{E}}$ metric verifies the correspondence criterion, detail explanations are provided in Appendix 8.2. MFCC is a higher level of audio feature derived from the Mel-spectrogram, which can reflect audio description information. This metric evaluates how the descriptions of the midpoint morphed audio result in-between two end points (see l.142 - 144, the definition of the correspondence criterion).

---

### Author Response · Authors · 2024-11-14

We thank for the time and effort the reviewers invested in reviewing our paper. We greatly appreciate your insightful suggestions, many of which will help us to improve our work. We hope the following clarifications address some of your major concerns.

**1. Regarding contribution and novelty**

Our method is inspired by image morphing method; However, it is not a simply straightforward adaptation but with insights about domain knowledge of sound morphing. As we claimed in Section 3.1, sound morphing has its inherent nature, which is different from image.
To the best of our knowledge, SoundMorpher is the first open-world sound morphing method based on a diffusion model, and we already showcased its advantages in various real-world applications in our experiment.

**2. Regarding three criterions that similar to IMPUS**

As we claimed in l.67-68, 141-142 and 533-534, **the three criterions are proposed by Caetano \& Osaka (2012)**, in which motivated to provide a formal evaluation reference for sound morphing methods. However, Caetano \& Osaka (2012) didn't offer detail quantitative metrics, our contribution is to adapt a series of objective quantitative metrics according to the criterions proposed by Caetano \& Osaka (2012) and further provide a formal comprehensive evaluation system for future sound morphing studies.

**3. Regarding the lacking of baseline comparison**

Even though SMT is an out-of-date timbral morphing method; however, we can't find any other open-sourced related methods to make a direct comparison for the task of timbral morphing. The only open-sourced piano timbral morphing method (Tan et al. (2020)) requires additional MIDI notes, as we already claimed the difference in related work section.

For environmental sound morphing, we tried our best to make a comparison with the concurrent method (i.e., MorphFader) based on their demo samples, since this method hasn't released their codes.

If the reviewers have any suggestions about the baseline for a fair comparison, please elaborate.

**4. Regarding the comparison with the concurrent work (i.e., MorphFader)**

We have already provided a detail explanation for this comparison in Appendix 10 as well as spectra comparison in Figure 5. Since this experiment compares to a **concurrent work**, a slight improvement already showcases our superior.

---

### Note · Authors · 2024-12-16

I have read and agree with the venue's withdrawal policy on behalf of myself and my co-authors.